# Predicting mammalian hosts in which novel coronaviruses can be generated

Maya Wardeh [1,2 ✉], Matthew Baylis [1,3] & Marcus S. C. Blagrove [4 ✉]

Novel pathogenic coronaviruses – such as SARS-CoV and probably SARS-CoV-2 – arise by homologous recombination between co-infecting viruses in a single cell. Identifying possible sources of novel coronaviruses therefore requires identifying hosts of multiple coronaviruses; however, most coronavirus-host interactions remain unknown. Here, by deploying a meta-ensemble of similarity learners from three complementary perspectives (viral, mammalian and network), we predict which mammals are hosts of multiple coronaviruses. We predict that there are 11.5-fold more coronavirus-host associations, over 30-fold more potential SARS-CoV-2 recombination hosts, and over 40-fold more host species with four or more different subgenera of coronaviruses than have been observed to date at >0.5 mean probability cut-off (2.4-, 4.25- and 9-fold, respectively, at >0.9821). Our results demonstrate the large underappreciation of the potential scale of novel coronavirus generation in wild and domesticated animals. We identify high-risk species for coronavirus surveillance.

[1] Department of Livestock and One Health, Institute of Infection, Veterinary & Ecological Sciences, University of Liverpool, Liverpool, UK. [2] Department of Mathematical Sciences, University of Liverpool, Liverpool, UK. [3] Health Protection Research Unit in Emerging and Zoonotic Infections, University of Liverpool, Liverpool, UK. [4] Department of Evolution, Ecology and Behaviour, Institute of Infection, Veterinary & Ecological Sciences, University of Liverpool, Liverpool, UK. ✉email: maya.wardeh@liverpool.ac.uk; marcus.blagrove@liverpool.ac.uk

The generation and emergence of three novel respiratory coronaviruses from mammalian reservoirs into human populations in the last 20 years, including one which has achieved pandemic status, suggests that one of the most pressing current research questions is: in which reservoirs could the next novel coronaviruses be generated and emerge from in future? Armed with this knowledge, we may be able to reduce the chance of emergence into human populations, such as by the strict monitoring and enforced separation of the identified hosts, in live animal markets, farms, and other close-quarters environments; or we may be able to develop potential mitigations in advance.

Coronaviridae are a family of positive sense RNA viruses, which can cause an array of diseases. In humans, these range from mild cold-like illnesses to lethal respiratory tract infections. Seven coronaviruses are known to infect humans[1], SARS-CoV, MERS-CoV and SARS-CoV-2 causing severe disease, while HKU1, NL63, OC43 and 229E tend towards milder symptoms in most patients[2].

Coronaviruses undergo frequent host-shifting events between non-human animal species, or non-human animals and humans[3–5], a process that may involve changes to the cells or tissues that the viruses infect (virus tropism). Such shifts have resulted in new animal diseases (such as bovine coronavirus disease[6] and canine coronavirus disease[7]), and human diseases (such as OC43[8] and 229E[9]). The aetiological agent of COVID-19, SARS-CoV-2, is proposed to have originated in bats[10] and shifted to humans via an intermediate reservoir host, likely a species of pangolin[11].

Comparison of the genetic sequences of bat and human coronaviruses has revealed five potentially important genetic regions involved in host specificity and shifting, with the Spike receptor binding domain believed to be the most important[3,12]. Homologous recombination is a natural process, which brings together new combinations of genetic material, and hence new viral strains, from two similar non-identical parent strains of virus. This recombination occurs when different strains co-infect an individual animal, with sequences from each parent strain in the genetic make-up of progeny virus. Homologous recombination has previously been demonstrated in many important viruses such as human immunodeficiency virus (HIV)[13], classical swine fever virus[14] and throughout the Coronaviridae[12,15]. Homologous recombination in Spike has been implicated in the generation of SARS-CoV-2[15], although investigations are still ongoing.

As well as instigating host-shifting, homologous recombination in other regions of the virus genome could also introduce novel phenotypes to coronavirus strains already infectious to humans. There are at least seven potential regions for homologous recombination in the replicase and Spike regions of the SARS-CoV genome alone, with possible recombination partner viruses from a range of other mammalian and human coronaviruses[16]. Recombination events between two compatible partner strains in a shared host could thus lead to future novel coronaviruses, either by enabling pre-existing mammalian strains to infect humans, or by adding new phenotypes arising from different alleles to pre-existing human-affecting strains.

The most fundamental requirement for homologous recombination to take place is the co-infection of a single host with multiple coronaviruses. However, our understanding of which hosts are permissive to which coronaviruses, the prerequisite to identifying which hosts are potential sites for this recombination (henceforth termed 'recombination hosts'), remains extremely limited. Here, we utilise a similarity-based machine-learning pipeline to address this significant knowledge gap. Our approach predicts associations between coronaviruses and their potential mammalian hosts by integrating three perspectives or points of view encompassing: (1) genomic features depicting different aspects of coronaviruses (e.g., secondary structure, codon usage bias) extracted from complete genomes (sequences = 3271, virus strains = 411); (2) ecological, phylogenetic and geospatial traits of potential mammalian hosts ($n = 876$); and (3) characteristics of the network that describes the linkage of coronaviruses to their observed hosts, which expresses our current knowledge of sharing of coronaviruses between various hosts and host groups.

Topological features of ecological networks have been successfully utilised to enhance our understanding of pathogen sharing[17,18], disease emergence and spill-over events[19], and as means to predict missing links in host–pathogen networks[20–22]. Here, we capture this topology, and relations between coronaviruses and hosts in our network, by means of node (coronaviruses and hosts) embeddings using DeepWalk[23]—a deep learning method that has been successfully used to predict drug-target[24] and lncRNA-disease associations[25].

Our pipeline transforms the above features into similarities (between viruses and between hosts) and uses them to give scores to virus–mammal associations indicating how likely they are to occur. Our framework then ensembles its constituent learners to produce testable predictions of mammalian hosts of multiple coronaviruses, in order to answer the following questions: (1) which species may be unidentified mammalian reservoirs of coronaviruses? (2) What are the most probable mammalian host species in which coronavirus homologous recombination could occur? And (3) which coronaviruses are most likely to co-infect hosts, and thus act as sources for future novel viruses?

In the following work, we deploy a meta-ensemble of similarity learners from the three complementary perspectives (viral, mammalian and network) and use it to predict which mammals are hosts of multiple coronaviruses. Using this pipeline, we demonstrate that there is currently a large underappreciation of the potential scale of novel coronavirus generation in wild and domesticated animals. Specifically, we predict there are 11.5-fold more coronavirus–host associations, over 30-fold more potential SARS-CoV-2 recombination hosts, and over 40-fold more host species with four or more different subgenera of coronaviruses than have been observed to date at >0.5 mean probability cut-off (2.4-, 4.25- and 9-fold, respectively, at >0.9821). We use these data to identify potential high-risk species, which we recommend for coronavirus surveillance.

## Results

**Predicted recombination hosts of SARS-CoV-2.** Our pipeline to predict associations between coronaviruses and their mammalian hosts indicated a total of 126 non-human species in which SARS-CoV-2 could be found, mean probability cut-off > 0.5, when subtracting (adding) standard deviation (SD) from the mean the number of predicted hosts is 85 (169). For simplicity, we report SD hereafter as −/+ from predicted values at reported probability cut-offs, here: SD = −41/+43. The number of predicted SARS-CoV-2 associations at cut-offs >0.75 and ≥0.9821 was: 103 (−40/+141) and 17 (−8/+126), respectively. The breakdown of these hosts by order is shown in Table 1. Figure 1 illustrates these predicted hosts, the probability of their association with SARS-COV-2, as well as numbers of known and unobserved (predicted) coronaviruses that could be found in each potential reservoir of SARS-CoV-2 (Supplementary Data 1 lists full predictions).

**Summary of predictions for all coronaviruses.** Overall, our pipeline predicted 4438 (SD = −1903/+2256, cut-off > 0.5) previously unobserved associations that potentially exist between 300 (SD = 0/+3) mammals and 204 coronaviruses (species or strains, SD = −60/+13). The number of unobserved associations at probability cut-offs >0.75 and ≥0.9821 was: 3087 (−1747/+2391)

**Table 1 Observed and predicted number of hosts of SARS-CoV-2 (by mammalian order), and observed and predicted number of hosts with ten or more coronaviruses (from our set of 411 species or strains).**

| Mammalian order | Observed and predicted hosts of SARS-CoV-2 | | | Observed and predicted hosts with ten or more coronaviruses | | |
|---|---|---|---|---|---|---|
| | Cut-off > 0.5 | Cut-off > 0.75 | Cut-off ≥ 0.9821 | Cut-off > 0.5 | Cut-off > 0.75 | Cut-off ≥ 0.9821 |
| Artiodactyla | 18 (−8/+0) | 15 (−11/+3) | 0 (0/+18) | 20 (−2/+0) | 20 (−13/+0) | 3 (−2/+17) |
| Carnivora | 37 (0/+0) | 37 (−14/+0) | 12 (−5/+25) | 35 (−22/+2) | 20 (−14/+17) | 3 (−2/+30) |
| Chiroptera | 25 (−19/+0) | 13 (−12/+12) | 1 (−1/+24) | 129 (−87/+53) | 56 (−38/+105) | 7 (−3/+102) |
| Eulipotyphla | 5 (−4/+0) | 3 (−3/+2) | 0 (0/+5) | 5 (0/+0) | 5 (−4/+0) | 0 (0/+2) |
| Lagomorpha | 2 (−1/+0) | 2 (−2/+0) | 0 (0/+2) | 2 (0/+0) | 2 (−1/+0) | 0 (0/+1) |
| Perissodactyla | 2 (0/+0) | 2 (−2/+0) | 0 (0/+2) | 2 (0/+0) | 2 (−2/+0) | 0 (0/+2) |
| Pholidota | 1 (0/+0) | 1 (−1/+0) | 0 (0/+1) | 1 (0/+0) | 1 (−1/+0) | 0 (0/+1) |
| Primates (non-human) | 4 (0/+0) | 4 (−1/+0) | 0 (0/+4) | 4 (0/+0) | 4 (−4/+0) | 0 (0/+3) |
| Rodentia (excluding laboratory species) | 32 (−9/+0) | 26 (−17/+6) | 4 (−3/+28) | 33 (−4/+3) | 30 (−27/+6) | 0 (0/+4) |

Numbers are presented at three probability cut-offs: >0.5, >0/75 and ≥0.9. Values in brackets are generated by subtracting/adding standard deviation (SD) from the mean probability of the ensemble (100 runs) and generating predictions at the listed cut-offs.

between 300 (−16/+0) mammals and 181 (−127/+26) coronaviruses and 601 (−412/+3723) between 224 (−91/+76) mammals and 31 (−7/+171) coronaviruses, respectively. Our model predicts there are 115 (0/+3) [115 (−4/+0), 96 (−31/+19), at cut-offs > 0.75 and ≥0.9821] mammalian species with no previously observed associations with the 411 input viruses (hereafter we display results derived from >0.5 cut-off; results obtained at >0.75 and ≥0.9821 cut-offs are presented in square brackets).

On average, each coronavirus (species or strain, complete genome available, $n = 411$) is predicted to have 12.56 (−4.92/+5.83) mammalian hosts [9.06 (−4.51/+6.18); 2.64 (−1.06/+9.62)]. Similarly, each mammalian species ($n = 876$, known hosts = 185, predicted hosts = 300 (−0/+3) [300 (−4/+0); 281 (−0/+19)]) is host to, on average, 5.55 (−2.17/+2.58) coronaviruses [9.06 (−4.51/+6.18); 1.17 (−0.47/+4.25)]. Supplementary Data 2 and 3 provide results for coronaviruses and mammalian hosts, respectively.

Figure 2 presents 50 potential mammalian recombination hosts of coronaviruses. Our model predicts 231 (−115/+58) [140 (−104/+128); 13 (−7/+217)] mammalian species (excluding humans and lab rodents) that could host 10 or more of the 411 coronavirus species or strains for which complete genome sequences were available. The breakdowns of these hosts by order are shown in Table 1.

**Coronavirus–mammalian networks**. The addition of predicted associations increased the diversity (mean phylogenetic distance) of mammalian hosts per coronavirus, as well as the diversity (mean genetic distance) of coronaviruses per mammalian host (Table 2 lists these changes). Furthermore, we captured the changes in structure of the bipartite network linking coronaviruses with their mammalian hosts (Fig. 3). On the one hand, the nestedness of the network increased (ranging from: 4.06-fold at 0.9821 to 10.17-fold at 0.50 cut-off, Table 2). On the other hand, the non-independence (checkerboard score (C-score)) of coronaviruses and mammalian hosts decreased with the addition of new links. Larger values of C-score suggest viral and host communities have little or no overlap in host or virus preferences (e.g., tendencies of coronavirus types to be clustered amongst certain host communities), as visualised in Fig. 3.

**Validation**. We validated our analytical pipeline externally against 20 held-out test sets (as described in method section below). On average, our GBM ensemble achieved AUC = 0.948

(±0.029 SD), 0.944 (±0.024), 0.843 (±0.045); true skill statistics (TSS) = 0.832 (±0.057), 0.887 (±0.048), 0.687 (±0.091); and F-score = 0.102 (±0.049), 0.141 (±0.055), 0.283 (±0.062), at probability cut-offs: >0.5, >0.75 and ≥0.9821, respectively (Supplementary Figs. 7–12).

**Discussion**

In this study, we deployed a meta-ensemble of similarity learners from three complementary perspectives (viral, mammalian and network), to predict the occurrence of associations between 411 known coronaviruses and 876 mammal species. We predict 11.54-fold increase—prediction cut-off > 0.5 [8.33-fold, 2.43-fold, cut-offs >0.75, ≥0.9821, respectively, cut-offs presented in this format hereafter], leading to the prediction that there are many more mammalian species than are currently known in which more than one coronavirus can occur. These hosts of multiple coronaviruses are potential sources of new coronavirus strains by homologous recombination. Here, we discuss the large number of candidate hosts in which homologous recombination of coronaviruses could result in the generation of novel pathogenic strains, as well as the substantial underestimation of the range of viruses which could recombine based on observed data. Our results are also discussed in terms of which host species are high priority targets for surveillance, both short and long term.

Give that coronaviruses frequently undergo homologous recombination when they co-infect a host, and that SARS-CoV-2 is highly infectious to humans, the most immediate threat to public health is recombination of other coronaviruses with SARS-CoV-2. Such recombination could readily produce further novel viruses with both the infectivity of SARS-CoV-2 and additional pathogenicity or viral tropism from elsewhere in the Coronaviridae. (See Supplementary Data 1 and 3 for comprehensive list of mammals predicted to be hosts of SARS-CoV-2 as well as several other coronaviruses).

Taking only observed data, there are four non-human mammalian hosts known to associate with SARS-CoV-2 and at least one other coronavirus, and a total of 504 different unique interactions between SARS-CoV-2 and other coronaviruses (counting all combinations of virus and host individually). Any of these SARS-CoV-2 hosts that are also hosts of other coronaviruses are potential recombination hosts in which novel coronaviruses derived from SARS-CoV-2 could be generated in the future. However, when we add in our model's predicted interactions this becomes 126 SARS-CoV-2 hosts and 2544 total unique interactions [103 hosts and 1898 unique interactions; 17

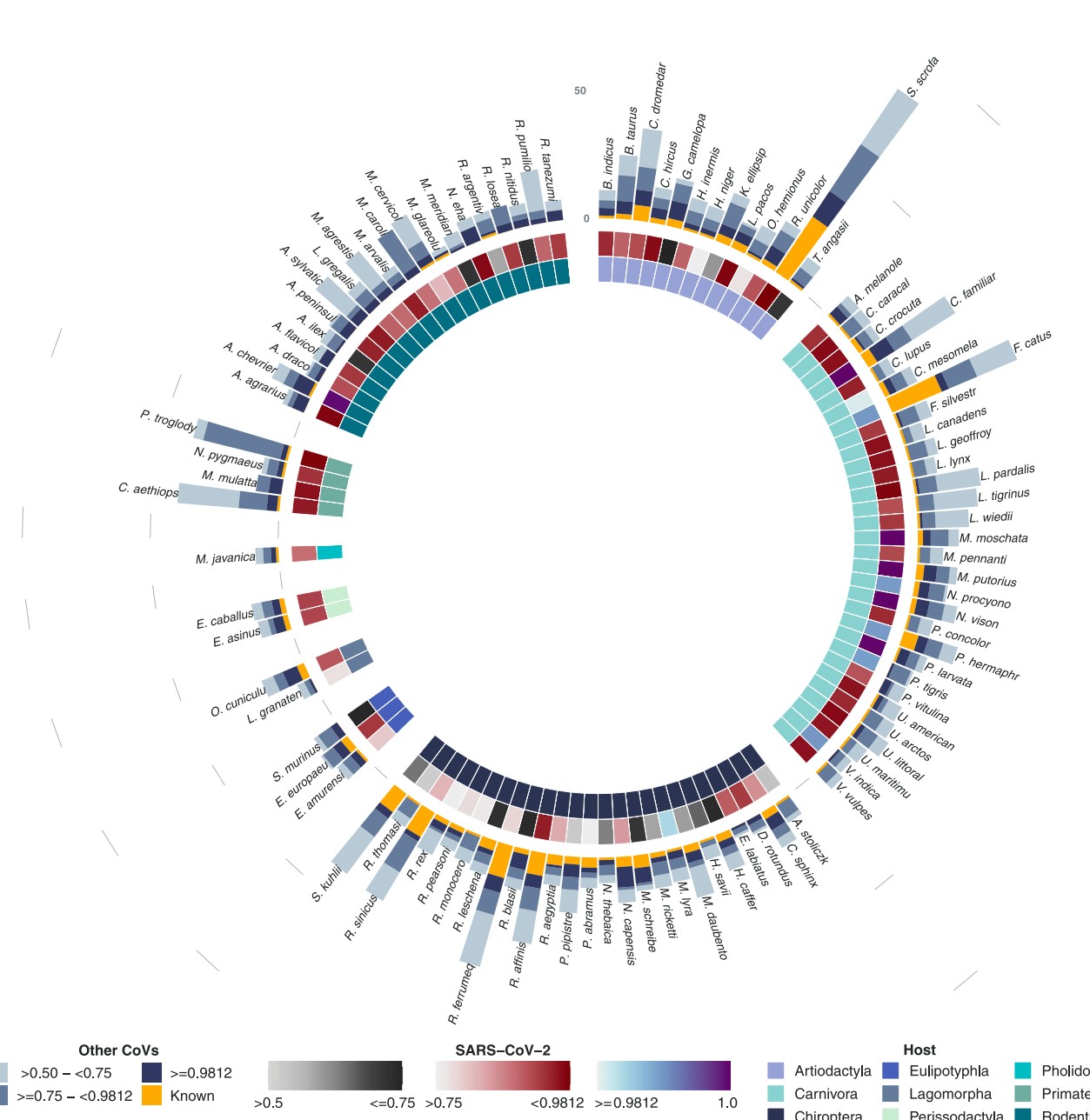

**Fig. 1 Model predictions for potential hosts of SARS-Cov-2.** Predicted hosts are grouped by order (inner circle). Middle circle presents probability of association between host and SARS-CoV-2 (grey scale indicates predicted associations with probability in range > 0.5 to ≤0.75. Red scale indicates predicted associations with probability in range > 0.75 to <0.9821. Blue to purple scale present indicates associations with probability ≥ 0.9821). Yellow bars represent number of coronaviruses (species or strains) observed to be found in each host. Blue stacked bars represent other coronaviruses predicted to be found in each host by our model. Predicted coronaviruses per host are grouped by prediction probability into three categories (from inside to outside): ≥0.9821, >0.75 to <0.9821 and >0.5 to ≤0.75. Results for humans and lab rodents are not shown to prevent the scale from contracting and making other comparisons difficult. Supplementary Fig. 14 illustrates full results including these hosts. Full results are listed in Supplementary Data 1.

hosts and 563 interactions]; indicating that observed data are missing 31.5-fold of the total number of predicted recombination hosts [25.75-fold; 4.25-fold], and 5.05-fold increase [3.77-fold; 1.12-fold] of the predicted unique associations. These large-fold increases in the number of predicted hosts and associations demonstrate that the potential for homologous recombination between SARS-CoV-2 and other coronaviruses, which could lead to new pathogenic strains, is highly underestimated, both in terms

of the range of hosts as well as the number of interactions within known hosts.

Our model has successfully highlighted known important recombination hosts of coronaviruses, adding confidence to our methodology. The Asian palm civet (*Paradoxurus hermaphroditus*), a viverrid native to south and southeast Asia, was predicted by our model as a potential host of 32 [26; 10] different coronaviruses (in addition to SARS-CoV-2) (vs. 6 observed). Genetic

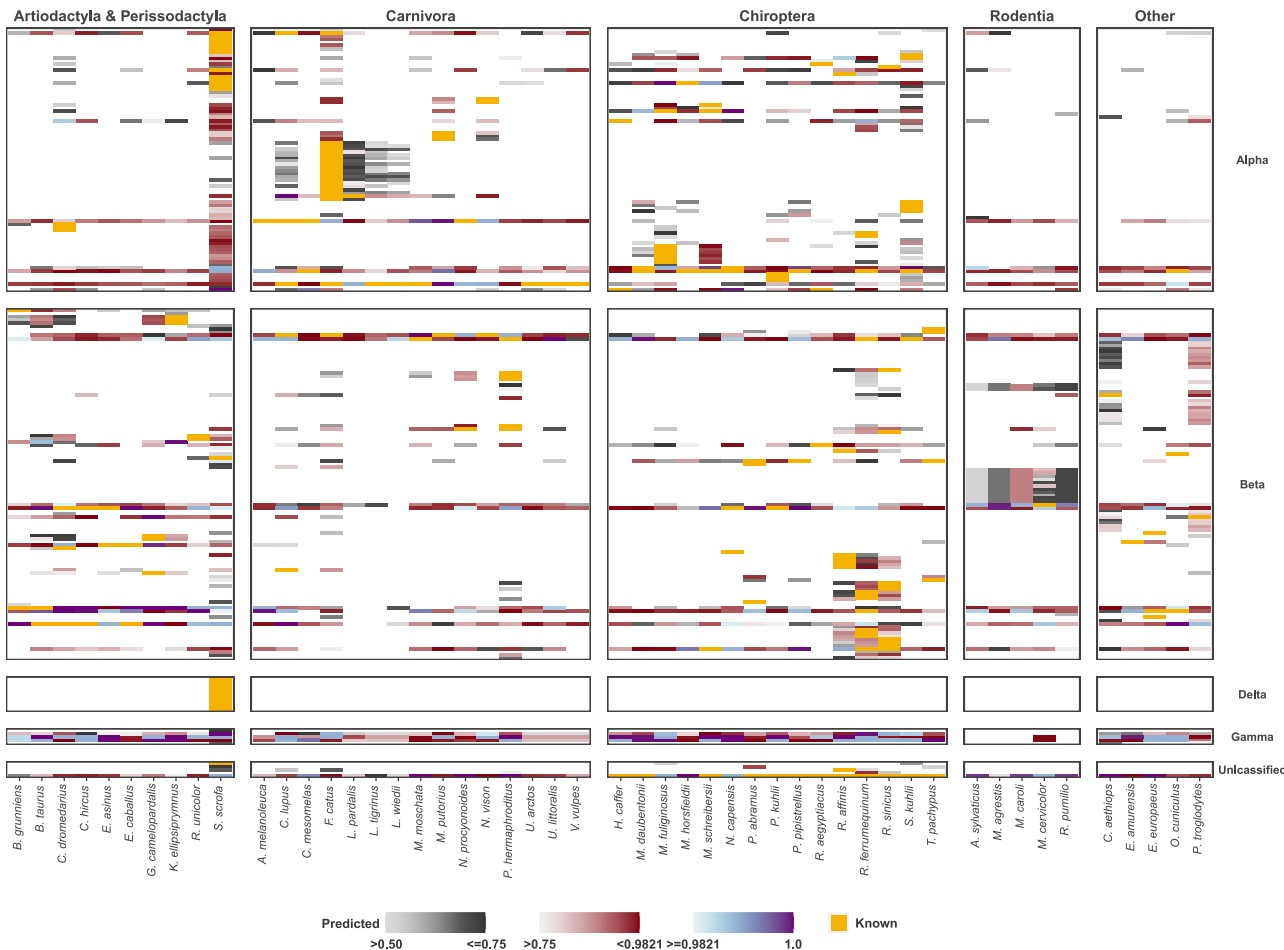

**Fig. 2 Observed and predicted mammalian hosts for coronaviruses.** Columns present mammalian hosts in four categories: Artiodactyla and Perissodactyla (top 10 hosts by number of predicted coronaviruses that could be found in each host), Carnivora (top 15 hosts), Chiroptera (top 15 hosts), Rodentia (top 5 hosts) and others (top 5 hosts). Rows present viruses ordered into five taxonomic groups: alphacoronaviruses, betacoronaviruses, deltacoronaviruses, gammacoronaviruses and unclassified Coronavirinae. Yellow cells represent observed associations between the host and the coronavirus. Grey/red/blue cells indicate the probability of predicted associations in three increasing probability ranges. White cells indicate no known or predicted association between host and virus (beneath cut-off probability of 0.5). Supplementary Data 4 lists full results. These results exclude humans and lab rodents. Supplementary Data 5 lists predictions for humans. Supplementary Fig. 15 illustrates full results including these hosts.

evolution analysis has shown that SARS-CoV-2 is closely related to coronaviruses derived from *P. hermaphroditus*[26] and has also highlighted its role as a reservoir for SARS-CoV[27], strongly supporting our findings that it is an important host in coronavirus recombination. This, together with the close association of *P. hermaphroditus*[26] with humans, for example, via bushmeat and the pet trade[28] and in 'battery cages' for the production of Kopi luwak coffee, highlights both the ability and opportunity of this species to act as a recombination host, with significantly more coronaviruses than have been observed. Furthermore, our model highlights both the greater horseshoe bat (*Rhinolophus ferrumequinum*), which is a known recombination host of SARS-CoV[29,30], as well as the intermediate horseshoe bat (*Rhinolophus affinis*), which is believed to be recombination host of SARS-CoV-2[10,31]. Our model predicts *R. ferrumequinum* to be a host to 68 [47; 19] different coronaviruses (including SARS-CoV-2) (vs. 13 observed); and for *R. affinis* to host 45 [32; 14] (vs. 9 observed). Our model also highlights the pangolin (*Manis javanica*), a suspected intermediate host for SARS-CoV-2[11] as a predicted host of an additional 14 [11; 2] different coronaviruses (vs. 1 observed).

The successful highlighting of speculated hosts for SARS-CoV and SARS-CoV-2 homologous recombination adds substantial confidence that our model is identifying the most important potential recombination hosts. Furthermore, our results suggest that the number of viruses that could potentially recombine even within these known hosts has been significantly under-ascertained, indicating that there still remains significant potential for further novel coronavirus generation in future from current known recombination hosts.

Our pipeline also identifies a diverse range of species not yet associated with SARS-CoV-2 recombination, but which are both predicted to host SARS-CoV-2 and other coronaviruses. These hosts represent new targets for surveillance of novel human pathogenic coronaviruses. Amongst the highest priority is the lesser Asiatic yellow bat (*Scotophilus kuhlii*), a known coronavirus host[32], common in east Asia but not well studied, and which features prominently with a large number of predicted interactions (48 [29; 12]). Our results also implicate the common hedgehog (*Erinaceus europaeus*), the European rabbit (*Oryctolagus cuniculus*) and the domestic cat (*Felis catus*) as predicted hosts for SARS-CoV-2 (confirmed for the cat[33]) and large numbers of other coronaviruses (20, 23, 65 [19, 18, 48; 7, 9, 24], for the hedgehog, rabbit and cat, respectively). The hedgehog and rabbit have previously been confirmed as hosts for other betacoronaviruses[34,35], which have no appreciable significance to

| Table 2 Bipartite network metrics calculated for original and predicted networks at three probability cut-offs: ≥0.9821, >0.75, and >0.50. | | | | |
|---|---|---|---|---|
| Metric | Original network | Cut-off ≥ 0.9821 | Cut-off > 0.75 | Cut-off > 0.5 |
| Mammalian diversity per virus | 0.057 | 0.093 (0.085–0.510)/1.632-fold (1.491–8.947) | 0.543 (0.131–0.548)/9.526-fold (2.298–9.614) | 0.519 (0.331–0.595)/9.11-fold (5.81–10.44) |
| Viral diversity per mammal | 0.241 | 0.501 (0.270–0.730)/2.079-fold (1.120–3.029) | 0.731 (0.666–0.729)/3.033-fold (2.763–3.025) | 0.73 (0.73–0.72)/3.03-fold (3.03–2.99) |
| Nestedness (NODF) | 6.065 | 24.599 (12.979–61.513)/4.056-fold (2.14–10.142) | 56.856 (38.165–65.653)/9.374-fold (6.293–10.825) | 61.705 (51.99–68.843)/10.17-fold (8.57–11.35) |
| C-score (CoVs) | 0.811 | 0.56 (0.666–0.125)/0.69-fold (0.821–0.154) | 0.165 (0.381–0.117)/0.203-fold (0.47–0.144) | 0.124 (0.213–0.104)/0.15-fold (0.26–0.13) |
| C-score (mammals) | 0.931 | 0.877 (0.92–0.53)/0.942-fold (0.988–0.569) | 0.61 (0.81–0.437)/0.655-fold (0.87–0.469) | 0.531 (0.689–0.378)/0.57-fold (0.74–0.41) |

human health. Our prediction of these species' potential inter-action with SARS-CoV-2 and considerable numbers of other coronaviruses, as well as the latter three species' close association to humans, identify them as high priority underestimated risks. In addition to these human-associated species, both the chimpanzee (*Pan troglodytes*) and African green monkey (*Chlorocebus aethiops*) have large numbers of predicted associations (51, 46 [47, 22; 3, 4]), and given their relatedness to humans and their importance in the emergence of viruses such as DENV[36] and HIV[37], also serve as other high priority species for surveillance.

The most prominent result for a SARS-CoV-2 recombination host is the domestic pig (*Sus scrofa*), having the most predicted associations of all included non-human mammals (121 [95; 38] additional coronaviruses). The pig is a major known mammalian coronavirus host, harbouring both a large number (26) of observed coronaviruses, as well as a wide diversity (listed in Supplementary Data 4). Given the large number of predicted viral associations presented here, the pig's close association to humans, its known reservoir status for many other zoonotic viruses, and its involvement in genetic recombination of some of these viruses[38], the pig is predicted to be one of the foremost candidates an important recombination host.

As an example of the utilisation of our model from the per-spective of likely future viral homologous recombination events, Banerjee et al.[39] bioinformatically identified potential genomic regions of homologous recombination between MERS-CoV and SARS-CoV-2. They highlighted a significant risk of the highly human-to-human transmissible SARS-CoV-2 acquiring the con-siderably more pathogenic (i.e., in terms of case-fatality rate) phenotypes of MERS-CoV. The work presented here identifies 102 [75; 4] potential recombination hosts (excluding humans and laboratory rodents) of the two viruses. Together, our work and Banerjee et al.[39], we provide evidence for both the possible pro-duction of a potentially severe future recombinant coronavirus and identify the hosts in which this threat is most likely to be generated (see Supplementary Data 6). We recommend mon-itoring for this event.

Alongside the more immediate threat of homologous recom-bination directly with SARS-CoV-2, we also present our predicted associations between all mammals and all coronaviruses. These associations represent the longer-term potential for background viral evolution via homologous recombination in all species. These data also indicate that there is a 11.54-fold underestimation in the number of associations, with 421 observed associations and 4438 predicted [3087 (8.33-fold); 601 (2.43-fold)]. This is visually represented in Fig. 3, which illustrates the bipartite network of virus and host for observed associations (A), and predicted associations (B–D); with a marked increase in connectivity between our mammalian hosts and coronaviruses, even at the most stringent probability cut-off. This indicates that the poten-tial for homologous recombination between coronaviruses is substantially underestimated using just observed data.

Furthermore, our model predicts that the associations between more diverse coronaviruses is also underestimated, for example, the number of included host species with four or more different subgenera of coronaviruses increases by 41.57-fold from 7 observed to 291 predicted [39.57-fold, 277; 9.00-fold, 63] (Table 2 shows the degree of diversity of coronaviruses in mammalian host species highlighted in Fig. 2). The high degree of potential co-infections including different subgenera and genera seen in our results emphasises the level of new genetic diversity possible via homologous recombination in these host species. A similar array of host species is highlighted for total associations as was seen for SARS-CoV-2 potential recombination hosts, including the common pig, the lesser Asiatic yellow bat, and both the greater and intermediate horseshoe bats, whilst notable additions

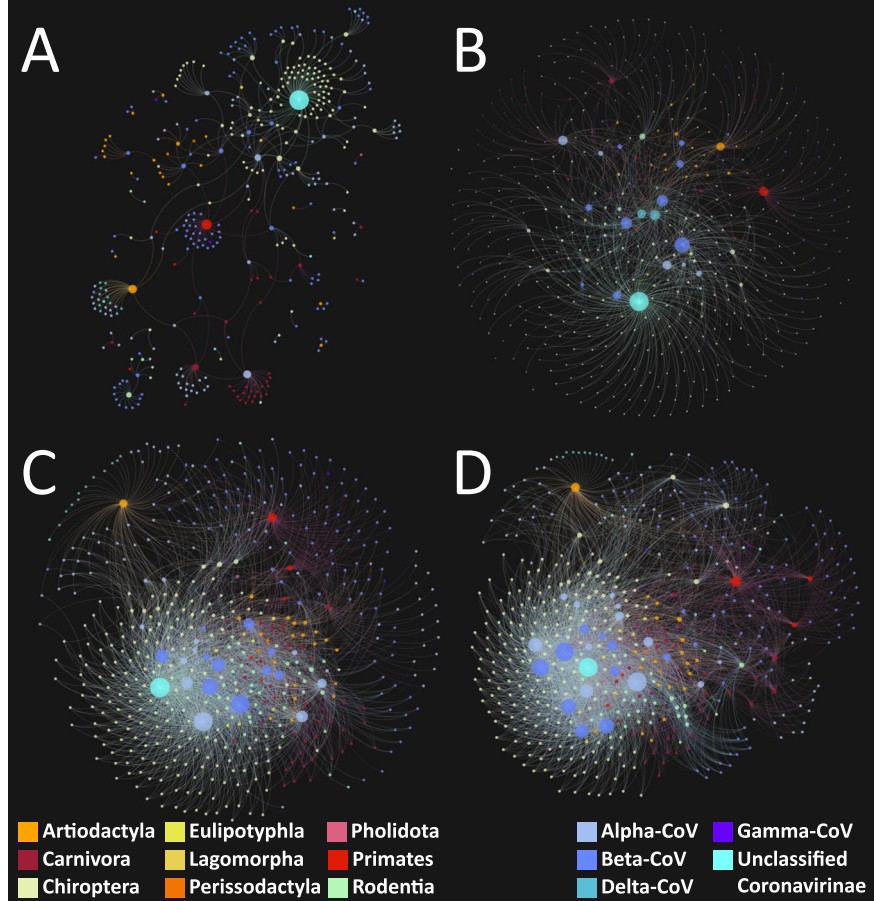

**Fig. 3 Bipartite networks linking coronaviruses with mammalian hosts.** Panel (**A**): original bipartite network based on known/observed virus–host associations extracted from meta-data accompanying genomic sequences and supplemented with publications data from the ENHanCEd Infectious Diseases Database (EID2). Panels (**B**–**D**) show predicted bipartite networks using our predicted virus–host associations at different cut-offs: ≥0.9821, >0.75 and >0.5, respectively, for mean probability of associations.

include the dromedary camel (*Camelus dromedaries*). The camel is a known host of multiple coronaviruses and the primary route of transmission of MERS-CoV to humans[40]. Our results suggest that monitoring for background viral evolution via homologous recombination would focus on a similar array of hosts, with a few additions, as monitoring for SARS-CoV-2 recombination. Again, our results strongly suggest that the potential array of viruses which could recombine in hosts is substantially underestimated, reinforcing the message that continued monitoring is essential.

Methodologically, the novelty of our approach lies in integrating three points of view: that of the coronaviruses, that of their potential mammalian hosts and that of the network summarising our knowledge to date of sharing of coronaviruses in their hosts. Additionally, the incorporation of similarity-based learners in our three-perspective approach enabled us to capture new hosts (i.e., with no known association with coronaviruses), thus avoiding a main limitation of approaches, which rely only on networks and their topology. By constructing a comprehensive set of similarity learners in each point of view and combining these learners non-linearly (via GBM meta-ensemble), a strength of our analytical pipeline is that it is able to predict potential recombination hosts of coronaviruses without any prerequisite knowledge or assumptions. Our method does not make assumptions about which parts of the coronavirus or host genomes are important, or integration of receptor (e.g., ACE2) information, or focusing on certain groups of hosts (e.g., bats or primates). This 'no-pre-conceptions' approach enables us to analyse without being

restricted by our current incomplete knowledge of the specific biological and molecular mechanisms, which govern host-virus permissibility. Current restrictions include lack of sequencing, annotation and expression analysis of receptor (e.g., ACE2) in the vast majority of hosts, uncertainty over the receptor(s) utilised by many coronaviruses and knowledge of other factors leading to successful replication once the virus has entered the host cell. Whilst some of these details are known for a very limited number of well-studied hosts and coronaviruses, they are not for the vast majority, consequently, a study aiming for breadth of understanding across all mammalian hosts and coronaviruses is unable to utilise these limited data. Despite our 'no-preconceptions' approach having this distinct advantage, it is also a limitation of the predictions. As discussed in the next section, our predictions are consequently reliant upon a more limited set of information due to the breadth of the work. Where some data are available for a small subset of coronaviruses or their hosts (e.g., pathogenicity, virus titre), these data are not useable in this study as they do not exist for the vast majority of hosts/viruses.

We acknowledge certain limitations in our methodology, primarily pertaining to current incomplete data sets in the rapidly developing but still understudied field:

(1) The inclusion only of coronaviruses for which complete genomes could be found limited the number of coronaviruses (species or strain) for which we could compute meaningful similarities, and therefore predict potential

hosts. The same applies for our mammalian species—we only included mammalian hosts for which phylogenetic, ecological and geospatial data were available. As more data on sequenced coronaviruses or mammals become available in future, our model can be re-run to further improve predictions, and to validate predictions from earlier iterations.

(2) Virological knowledge of understudied coronaviruses and their host interactions. For the vast majority of observed virus–host associations it is unknown if these hosts are natural, intermediate or 'dead-end' hosts. Also unknown are more clinical traits of the infections in the overwhelming majority of associations, such as: pathogenicity, likelihood of infection, virus titre during infection, duration of infection, etc. knowledge of all of these factors could greatly add to our ability to assess 'likeliness' of homologous recombination, however, the available data are too limited for a study with the breadth of interactions we characterise here, and hence were unable to be included.

(3) Research effort, centring mainly on coronaviruses found in humans and their domesticated animals, can lead to overestimation of the potential of coronaviruses to recombine in frequently studied mammals, such as lab rodents that were excluded from the results reported here (similar to previous work[17]), and significantly, domesticated pigs and cats that we have found to be important recombination host species of coronaviruses. We believe that this limitation is partially mitigated; first, methodologically, the effect of research effort has been limited by capturing similarities from our three points of view (virus, host and network) and multiple characteristics therein. And second, this mitigation shows that in our results as other 'overstudied' mammals, such as cows and sheep, were not highlight by our model, which is consistent with them being considered less important hosts of coronaviruses, and certain understudied bats were highlighted as major potential hosts; together, these indicate that research effort is not a substantial driver of our results.

Recent testing of potential mammalian hosts for their susceptibility to SARS-CoV-2 has confirmed a number of our predictions, for example: *Nyctereutes procyonoides*[41,42]; *Bovines* (e.g., *Bison bonasus, Bos taurus, Bos indicus, Bubalus bubalis*), *Capra hircus, Equus caballus, Lama (Vicugna) pacos, Manis javanica, Oryctolagus cuniculus, Panthera leo, Rousettus leschenaultii, Sus scrofa* and *Vulpes vulpes*[42]; *Chlorocebus aethiops, Neovison vison, Macaca mulatta* and *Rousettus aegyptiacus*[43]. While limited in number, these post hoc confirmations add confidence to our framework and its predictions. As more host screening is performed in future, it will enable further evaluation of our predictions.

To follow-on from this work, we are investigating coronavirus–host interactions in two separate directions. The first is to expand our host range to include avian species, therefore, including the full range of important coronavirus hosts, and to inform our model with a species-level contact network for all hosts (indicating likeliness of a direct interaction). This will give a broader overview of potential coronavirus associations. Second, we are focusing our predictions on studying a subset of clinically important associations in more depth. This will allow us to utilise more specific information such as receptor and clinical data on the viraemia, which are only currently available for well-studied interactions.

In this study, we provide evidence that the potential for homologous recombination in mammalian hosts of coronaviruses is highly underestimated. The potential ability of the large numbers of hosts presented here to be hosts of multiple coronaviruses, including SARS-CoV-2, could provide the capacity for homologous recombination and hence potential production of further novel coronaviruses. Our methods deployed a meta-ensemble of similarity learners from three complementary perspectives (viral, mammalian and network), to predict each potential coronavirus–mammal association.

The current consensus is that SARS-CoV-2 was generated by homologous recombination; originally derived from coronaviruses in bats[10] and then shifted to humans via an intermediate reservoir host, likely a species of pangolin[11]. Importantly, the lineage of SARS-CoV-2 was deduced only after the outbreak in humans. With the greater understanding of the extent of mammalian host reservoirs and the potential recombination hosts we identify here, a targeted surveillance programme is now possible which would allow for this generation to be observed as it is happening and before a major outbreak. Such information could help inform prevention and mitigation strategies and provide a vital early warning system for future novel coronaviruses.

## Methods

### Viruses and mammalian data

*Viral genomic data.* Complete sequences of coronaviruses were downloaded from Genbank[44]. Sequences labelled with the terms: 'vaccine', 'construct', 'vector', 'recombinant' were removed from the analyses. In addition, we removed those associated with experimental infections where possible. This resulted in a total of 3264 sequences for 411 coronavirus species or strains (i.e., viruses below species level on NCBI taxonomy tree). Of those, 88 were sequences of coronavirus species, and 307 sequences of strains (in 25 coronavirus species, with total number of species included = 92). Of our included species, six in total were unclassified Coronavirinae (unclassified coronaviruses).

*Selection of potential mammalian hosts of coronaviruses.* We processed meta-data accompanying all sequences (including partial sequences but excluding vaccination and experimental infections) of coronaviruses uploaded to GenBank to extract information on hosts (to species level) of these coronaviruses. We supplemented these data with species-level hosts of coronaviruses extracted from scientific publications via the ENHanCEd Infectious Diseases Database (EID2)[45]. This resulted in identification of 313 known terrestrial mammalian hosts of coronaviruses (regardless of whether a complete genome was available or not, $n = 185$ mammalian species for which an association with a coronavirus with complete genome was identified). We expanded this set of potential hosts by including terrestrial mammalian species in genera containing at least one known host of coronavirus, and which are known to host one or more other virus species (excluding coronaviruses, information of whether the host is associated with a virus were obtained from EID2). This results in total of 876 mammalian species which were selected.

### Quantification of viral similarities. We computed three types of similarities between each two viral genomes as summarised below.

*Biases and codon usage.* We calculated proportion of each nucleotide of the total coding sequence length. We computed dinucleotide and codon biases[46] and codon-pair bias, measured as the codon-pair score[46,47] in each of the above sequences. This enabled us to produce for each genome sequence ($n = 3264$) the following feature vectors: nucleotide bias, dinucleotide bias, codon biased and codon-pair bias.

*Secondary structure.* Following alignment of sequences (using AlignSeqs function in R package Decipher[48]), we predicted the secondary structure for each sequence using PredictHEC function in the R package Decipher[48]. We obtained both states (final prediction), and probability of secondary structures for each sequence. We then computed for each 1% of the genome length both the coverage (number of times a structure was predicted) and mean probability of the structure (in the per cent of the genome considered). This enabled us to generate six vectors (length = 100) for each genome representing: mean probability and coverage for each of three possible structures—Helix (H), Beta-Sheet (E) or Coil (C).

*Genome dissimilarity (distance).* We calculated pairwise dissimilarity (in effect a hamming distance) between each two sequences in our set using the function DistanceMatrix in the R package Decipher[48]. We set this function to penalise gap-to-gap and gap-to-letter mismatches.

**Table 3 Mammalian phylogenetic, ecological and geospatial similarities.**

| Category | Similarities | Calculation | Reason for inclusion |
|---|---|---|---|
| Phylogeny | Phylogenetic | 1-normalised phylogenetic distance | Linked to sharing of viruses between mammals. |
| Ecology | Life-history and reproductive traits | Gower's distance matrices[59] | Life-history traits are a key feature in terms of metabolism and adaption to environment. Reproductive traits are potentially relevant in terms of within-host dynamics of viruses. |
| | Habitat utilisation | | Similar habitat utilisation might correlate with contact with similar viruses. |
| | Diet (proportional use of ten categories) | | Similar dietary habit might associate with similar viral assemblage. |
| Geospatial | Overlap (yes/no) | Intersection of species presence maps (Supplementary Note 1 and Supplementary Fig. 2) | Geographically overlapping host species tend to share viruses more often than geographically distant species. |
| | Climate | SNF (Similarity network fusion) of simplified climate similarity matrices (temperature and precipitation similarities) | Climate has been shown to influence a number of human and domestic mammal pathogens (including viruses). |
| | Geospatial | SNF of natural land cover, agriculture and farming, urbanisation and human population and mammalian diversity similarity matrices | Geospatial factors have been found to influence certain categories of host–pathogen associations (Supplementary Note 1). |

Pairwise similarities were calculated for our mammalian species ($n = 876$). Full details of these similarities, their sources and full justification are listed in Supplementary Note 1.

*Similarity quantification.* We transformed the feature (traits) vectors described above into similarities matrices between coronaviruses (species or strains). This was achieved by computing cosine similarity between these vectors in each category (e.g., codon-pair usage, H coverage, E probability). Formally, for each genomic feature ($n = 10$) presented by vector as described above, this similarity was calculated as follows:

$$\text{sim}_{\text{genomic}_l}(s_m, s_n) = \text{sim}_{\text{genomic}_l}(\mathbf{V}_m^{f_l}, \mathbf{V}_n^{f_l}) = \frac{\sum_{i=1}^d \left(\mathbf{V}_m^{f_l}[i] \times \mathbf{V}_n^{f_l}[i]\right)}{\sqrt{\sum_{i=1}^d \mathbf{V}_m^{f_l}[i]^2} \times \sqrt{\sum_{i=1}^d \mathbf{V}_n^{f_l}[i]^2}} \quad (1)$$

where $s_m$ and $s_n$ are two sequences presented by two feature vectors $\mathbf{V}_m^{f_l}$ and $\mathbf{V}_n^{f_l}$ from the genomic feature space $f_l$ (e.g., codon-pair bias) of the dimension d (e.g., $d = 100$ for H coverage).

We then calculated similarity between each pair of virus strains or species (in each category) as the mean of similarities between genomic sequences of the two virus strains or species (e.g., mean nucleotide bias similarity between all sequences of SARS-CoV-2 and all sequences of MERS-CoV presented the final nucleotide bias similarity between SARS-CoV-2 and MERS-CoV). This enabled us to generate 11 genomic features similarity matrices (the above 10 features represented by vectors and genomic similarity matrix) between our input coronaviruses. Supplementary Fig. 1 illustrates the process.

*Similarity network fusion (SNF).* We applied SNF[49] to integrate the following similarities in order to reduce our viral genomic feature space: (1) nucleotide, dinucleotide, codon and codon-pair usage biases were combined into one similarity matrix—genome bias similarity. And (2) Helix (H), Beta-Sheet (E) or Coil (C) mean probability and coverage similarities (six in total) were combined into one similarity matrix—secondary structure similarity.

SNF applies an iterative nonlinear method that updates every similarity matrix according to the other matrices via nearest neighbour approach and is scalable and is robust to noise and data heterogeneity. The integrated matrix captures both shared and complementary information from multiple similarities.

**Quantification of mammalian similarities.** We calculated a comprehensive set of mammalian similarities. Table 3 summarises these similarities and provides justification for inclusion. Supplementary Note 1 provides full details.

**Quantification of network similarities**

*Network construction.* We processed meta-data accompanying all sequences (including partial genome but excluding vaccination and experimental infections) of coronaviruses uploaded to Genbank[44] (accessed 4 May 2020) to extract information on hosts (to species level) of these coronaviruses. We supplemented these data with virus–host associations extracted from publications via the EID2 Database[45]. This resulted in 1669 associations between 1108 coronaviruses and 545 hosts (including non-mammalian hosts). We transformed these associations into a bipartite network linking species and strains of coronaviruses with their hosts.

*Quantification of topological features.* The above constructed network summarises our knowledge to date of associations between coronaviruses and their hosts, and

its topology expresses patterns of sharing these viruses between various hosts and host groups. Our analytical pipeline captures this topology, and relations between nodes in our network, by means of node embeddings. This approach encodes each node (here either a coronavirus or a host) with its own vector representation in a continuous vector space, which, in turn, enables us to calculate similarities between two nodes based on this representation.

We adopted DeepWalk[23] to compute vectorised representations for our coronaviruses and hosts from the network connecting them. DeepWalk[23] uses truncated random walks to get latent topological information of the network and obtains the vector representation of its nodes (in our case coronaviruses and their hosts) by maximising the probability of reaching a next node (i.e., probability of a virus–host association) given the previous nodes in these walks (Supplementary Note 2 lists further details).

*Similarity calculations.* Following the application of DeepWalk to compute the latent topological representation of our nodes, we calculated the similarity between two nodes in our network—n (vectorised as **N**) and m (vectorised as **M**), by using cosine similarity as follows[24,25]:

$$\text{sim}_{\text{network}}(n, m) = \text{sim}_{\text{network}}(\mathbf{M}, \mathbf{N}) = \frac{\sum_{i=1}^d (m_i \times n_i)}{\sqrt{\sum_{i=1}^d m_i^2} \times \sqrt{\sum_{i=1}^d n_i^2}} \quad (2)$$

where $d$ is the dimension of the vectorised representation of our nodes: **M**, **N**; and $m_i$ and $n_i$ are the components of vectors **M** and **N**, respectively.

**Similarity learning meta-ensemble—a multi-perspective approach.** Our analytical pipeline stacks 12 similarity learners into testable meta-ensembles. The constituent learners can be categorised by the following three 'points of view' (see also Supplementary Fig. 4 for a visual description):

*Coronaviruses—the virus point of view.* We assembled three models derived from (a) genome similarity, (b) genome biases and (c) genome secondary structure. Each of these learners gave each coronavirus–mammalian association ($v_i \rightarrow m_j$) a score, termed confidence, based on how similar the coronavirus $v_i$ is to known coronaviruses of mammalian species $m_j$, compared to how similar $v_i$ is to all included coronaviruses. In other words, if $v_i$ is more similar (e.g., based on genome secondary structure) to coronaviruses observed in host $m_j$ than it is similar to all coronaviruses (both observed in $m_j$ and not), then the association $v_i \rightarrow m_j$ is given a higher confidence score. Conversely, if $v_i$ is somewhat similar to coronaviruses observed in $m_j$, and also somewhat similar to viruses not known to infect this particular mammal, then the association $v_i \rightarrow m_j$ is given a medium confidence score. The association $v_i \rightarrow m_j$ is given a lower confidence score if $v_i$ is more similar to coronaviruses not known to infect $m_j$ than it is similar to coronaviruses observed in this host.

Formally, given an adjacency matrix A of dimensions $|\mathbf{V}| \times |\mathbf{M}|$ where $|\mathbf{V}|$ is number of coronaviruses included in this study (for which a complete genome could be found), and $|\mathbf{M}|$ is number of included mammals, such that for each $v_i \in \mathbf{V}$ and $m_j \in \mathbf{M}$, $a_{ij} = 1$ if an association is known to exist between the virus and the mammal, and 0 otherwise. Then for a similarity matrix $\text{sim}_{\text{viral}}$ corresponding to each of the similarity matrices calculated above, a learner from the viral point of

view is defined as follows[24,25]:

$$\text{confidence}_{\text{viral}}(v_i \rightarrow m_j) = \frac{\sum_{l=1,\,l\neq i}^{|\mathbf{V}|}(\text{sim}_{\text{viral}}(v_i,\,v_l) \times a_{lj})}{\sum_{l=1,\,l\neq i}^{|\mathbf{V}|}\text{sim}_{\text{viral}}(v_i,\,v_l)} \quad (3)$$

*Mammals—the host point of view.* We constructed seven learners from the similarities summarised in Table 3. Each of these learners calculated for every coronavirus–mammalian association $(v_i \rightarrow m_j)$ a confidence score based on how similar the mammalian species $m_j$ is to known hosts of the coronavirus $v_i$, compared to how similar $m_j$ is to mammals not associated with $v_i$. For instance, if $m_j$ is phylogenetically close to known hosts of $v_i$, and also phylogenetically distant to mammalian species not known to be associated with this coronavirus, then the phylogenetic similarly learner will assign $v_i \rightarrow m_j$ a higher confidence score. However, if $m_j$ does not overlap geographically with known hosts of $v_i$, then the geographical overlap learner will assign it a low (in effect 0) confidence score.

Formally, given the above-defined adjacency matrix **A**, and a similarity matrix $\text{sim}_{\text{mammalian}}$ corresponding to each of the similarity matrices summarised in Table 3, a learner from the mammalian point of view is defined as follows[24,25]:

$$\text{confidence}_{\text{mammalian}}(v_i \rightarrow m_j) = \frac{\sum_{l=1,\,l\neq j}^{|\mathbf{M}|}(\text{sim}_{\text{mammalian}}(m_j,\,m_l) \times a_{il})}{\sum_{l=1,\,l\neq j}^{|\mathbf{M}|}\text{sim}_{\text{mammalian}}(m_j,\,m_l)} \quad (4)$$

*Network—the network point of view.* We integrated two learners based on network similarities—one for mammals and one for coronaviruses. Formally, given the adjacency matrix **A**, our two learners from the network point of view as defined as follows[24]:

$$\text{confidence}_{\text{network}_V}(v_i \rightarrow m_j) = \frac{\sum_{l=1,\,l\neq i}^{|\mathbf{V}|}(\text{sim}_{\text{network}}(v_i,\,v_l) \times a_{lj})}{\sum_{l=1,\,l\neq i}^{|\mathbf{V}|}\text{sim}_{\text{network}}(v_i,\,v_l)} \quad (5)$$

$$\text{confidence}_{\text{network}_M}(v_i \rightarrow m_j) = \frac{\sum_{l=1,\,l\neq j}^{|\mathbf{M}|}(\text{sim}_{\text{network}}(m_j,\,m_l) \times a_{il})}{\sum_{l=1,\,l\neq j}^{|\mathbf{M}|}\text{sim}_{\text{network}}(m_j,\,m_l)} \quad (6)$$

*Ensemble construction.* We combined the learners described above by stacking them into ensembles (meta-ensembles) using Stochastic Gradient Boosting (GBM). The purpose of this combination is to incorporate the three points of views, as well as varied aspects of the coronaviruses and their mammalian potential hosts, into a generalisable, robust model[50]. We selected GBM as our stacking algorithm following an assessment of seven machine-learning algorithms using held-out test sets (20% of known associations randomly selected, $N = 5$—Supplementary Fig. 14). In addition, GBM is known for its ability to handle non-linearity and high-order interactions between constituent learners[51], and have been used to predict reservoirs of viruses[46] and zoonotic hot-spots[51]. We performed the training and optimisation (tuning) of these ensembles using the caret R Package[52].

*Sampling.* Our GBM ensembles comprised 100 replicate models. Each model was trained with balanced random samples using tenfold cross-validation (Supplementary Fig. 4). Final ensembles were generated by taking mean predictions (probability) of constituent models. Predictions were calculated form the mean probability at three cut-offs: >0.5 (standard), >0.75 and ≥0.9821. SD from mean probability was also generated and its values subtracted/added to predictions, to illustrate variation in the underlying replicate models.

*Validation and performance estimation.* We validated the performance of our analytical pipeline externally against 20 held-out test sets. Each test set was generated by splitting the set of observed associations between coronaviruses and their hosts into two random sets: a training set comprising 85% of all known associations and a test set comprising 15% of known associations. These test sets were held-out throughout the processes of generating similarity matrices; similarity learning, and assembling our learners, and were only used for the purposes of estimating performance metrics of our analytical pipeline. This resulted in 20 runs in which our ensemble learnt using only 85% of observed associations between our coronaviruses and their mammalian hosts. For each run, we calculated three performance metrics based on the mean probability across each set of 100 replicate models of the GBM meta-ensembles: AUC, true skill statistics (TSS) and F-score.

AUC is a threshold-independent measure of model predictive performance that is commonly used as a validation metric for host–pathogen predictive models[21,46]. Use of AUC has been criticised for its insensitivity to absolute predicted probability and its inclusion of a priori untenable prediction[51,53], and so we also calculated the TSS (TSS = sensitivity + specificity − 1)[54]. F-score captures the harmonic mean of the precision and recall and is often used with uneven class distribution. Our approach is relaxed with respect to false positives (unobserved associations), hence the low F-score recorded overall.

We selected three probability cut-offs for our meta-ensemble: 0.50, 0.75 and 0.9821. One extreme of our cut-off range (0.5) maximises the ability of our ensemble to detect known associations (higher AUC, lower F-score). The other (0.9821) is calculated so that 90% of known positives are captured by our ensemble, while reducing the number of additional associations predicted (higher F-score, lower AUC).

*Changes in network structure.* We quantified the diversity of the mammalian hosts of each coronavirus in our input by computing mean phylogenetic distance between these hosts. Similarly, we captured the diversity of coronaviruses associated with each mammalian species by calculating mean (hamming) distance between the genomes of these coronaviruses. We termed these two metrics: mammalian diversity per virus and viral diversity per mammal, respectively. We aggregated both metrics at the network level by means of simple average. This enabled us to quantify changes in these diversity metrics, at the level of network, with addition of predicted links at three probability cut-offs: >0.5, >0.75 and ≥0.9821.

In addition, we captured changes in the structure of the bipartite network linking CoVs with their mammalian hosts, with the addition of predicted associations, by computing a comprehensive set of structural properties (Supplementary Note 3) at the probability cut-offs mentioned above, and comparing the results with our original network. Here we ignore properties that deterministically change with the addition of links (e.g., degree centrality, connectance; Supplementary Table 2 lists all computed metrics and changes in their values). Instead, we focus on non-trivial structural properties. Specifically, we capture changes in network stability, by measuring its nestedness[55–57]; and we quantify non-independence in interaction patterns by means of C-score[58]. Supplementary Note 3 provides full definition of these concepts as well as other metrics we computed for our networks.

**Reporting summary.** Further information on research design is available in the Nature Research Reporting Summary linked to this article.

## Data availability
Genomic sequences of coronaviruses were obtained from NCBI GenBank, accession codes are listed in Supplementary Data 7. Coronaviruses–hosts associations were obtained from the ENHanCEd Infectious Diseases Database (EID2: https://eid2.liverpool.ac.uk/). Mammalian and geospatial data were obtained from open-access data sources. These sources are listed in detail, and their DOIs are provided in the Supplementary Information file. Data used can be found here: https://doi.org/10.6084/m9.figshare.13110896, with the exception of mammalian presence shapefiles and raw climate data (due to their large size)—these data can be obtained from the authors or directly from the sources listed in the Supplementary Information file.

## Code availability
All codes used in our analyses are made available via figshare (https://doi.org/10.6084/m9.figshare.13110896).

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

## Acknowledgements

M.W. acknowledges the support from BBSRC and MRC for the National Productivity Investment Fund (NPIF) fellowship (MR/R024898/1). M.W. and M.S.C.B acknowledge support from BBSRC IAA COVID - 168478. Establishment of the EID2 database was funded by a UK Research Council Grant (NE/G002827/1) to M.B., as part of an ERA-NET Environmental Health award to M.B.; subsequently, it has been further developed and maintained by BBSRC Tools and Resources Development Fund awards (BB/K003798/1; BB/N02320X/1) to M.B., and the National Institute for Health Research Health Protection Research Unit (NIHR HPRU) in Emerging and Zoonotic Infections at the University of Liverpool in partnership with Public Health England and Liverpool School of Tropical Medicine.

## Author contributions

Conceived and designed the study: M.W. and M.S.C.B. Compiled the data and designed and implemented analytical pipeline: M.W. Analysed and interpreted the data: M.W. and M.S.C.B. Established the EID2 database: M.W. and M.B. Wrote the paper: M.W., M.B. and M.S.C.B.

## Competing interests

The authors declare no competing interests.
