## [Peer Review File · Nature Communications]

Reviewers' Comments:

Reviewer #1:

Remarks to the Author:

In this manuscript, the authors use multiple similarity learners to predict the potential pool of mammalian hosts that are capable of supporting coronavirus infection. Using these approaches, the authors show that there are manifold more potential coronavirus-host associations and SARS-CoV-2 recombination hosts than have been currently characterized or described. These findings are of particular value considering the current pandemic.

Overall, the study is elegant, informative, and thorough, and the writing is strong. One comment that may be particularly germane to the current pandemic is that the authors did not include a comparison of ace2 expression vs. similarity to human ACE2 in their algorithms, which would greatly strengthen the argument for a largely uncharacterized pool of potential hosts of SARS-CoV-2. If this is planned for a future study, a discussion of its potential application is warranted.

Reviewer #3:

Remarks to the Author:

Wardeh et al provided a methodology for predicting mammalian hosts which are likely to carry coronaviruses. Recombination research is by far the most difficult to model in the field of viral evolution, because it can lead to uncertain judgements on host and phylogeny. I found the study design and the results presented are very novel and interesting. However several comments below should be addressed.

1. Zoonotic coronaviruses do recombine easily. However, please do not use a positive tone to conclude that SARS-CoV-2 is a recombinant virus, as investigations into its origin and natural hosts are clearly under way around the world. It's not easy to conduct this type of study, e.g. it took more than a decade to find out the origin of SARS-CoV.
2. How do authors define host types? As far as I am aware, coronaviruses could be isolated from non-natural hosts i.e. intermediate hosts, and the intermediate hosts are clearly not responsible for the origin of coronaviruses.
3. How did the authors select the best algorithm for the prediction?
3. Recombination could have occurred in any types of hosts and not just limited to a certain type of host. How did the authors define the different types of hosts in their study, i.e. natural hosts, intermediate hosts and dead-end hosts? Are there any literatures showing certain type of hosts are easily targeted by coronaviruses?
4. Given the receptors for most types of coronaviruses are unknown, it is a bit risky to 'predict' different coronaviruses shared by same/similar hosts. Some 'hosts' could be only transient hosts that hardly provide any evolution forces into the evolution of coronaviruses. Could the authors give a successful predicted case here?
5. Why there are no birds which are natural hosts for some gamma-coronaviruses?
6. Could the authors use different colors in Figure 3? It is a bit hard to see any pattern in this figure.
7. What are un-classified coronaviruses?

Reviewer #4:

Remarks to the Author:

In this paper, the authors use data on 411 coronavirus strains, 876 hosts and network of known host-pathogen interactions between these entities to predict host-pathogen interactions that may exist but have not yet been documented. I have a few comments on the study design and some minor corrections.

Firstly, I appreciate that the motivation of the study was to identify different coronaviruses that have the potential to recombine within the same host and thus gain the ability to infect humans, but it seems a shame to have not included humans in the host-pathogen network. Surely the potential for coronaviruses to coinfect and recombine within humans would also be relevant here, and understanding how we compare to these other hosts in terms of the diversity of coronaviruses that are predicted to infect us would be interesting. It may also give a better opportunity to test whether host range predictions are correct, assuming the diversity of viruses that infect humans is better studied than that of animals.

Secondly, the predicted expansion to the connectivity of the host-pathogen network is substantial. I would prefer the analysis to focus more on the predictions made at the 0.9 probability level than the 0.5, to add more confidence to the conclusions from the modelling side. In the Supplementary data, tests on held-out datasets 1 and 3 yield very different performance metrics than tests 2, 4 and 5 - can the authors comment on why this is? Also, for the accuracy assessments involving binary predictions, which probability cutoff was used? I would like to see confusion matrices as well as the metrics presented in Figure S6, to get a better feel for the raw numbers of true/false positives/negatives.

The ability to infect a host is one factor, but the amount of time a given virus spends in a given host, and the viral load in that host, should also matter in terms of the probability to two strains cohabiting a host and recombining. I would be interested to see the probabilities assigned here compared to any available data on the efficiency of CoVs to infect different hosts - are they correlated?

Did the individual components in the meta-ensemble agree or disagree about host-pathogen links? It wasn't clear whether multiple independent lines of evidence led to these conclusions, or whether a single source of support tended to be sufficient to draw a link.

Minor corrections:

Figure 1: the colouring of the number of coronaviruses predicted to be in each host may be a bit off here, as it looks like there are no predicted links with ≥ 0.9 support

SARS-CoV-2 is at one point written as SARS-COV2 - please make these consistent

Line 312 - some spelling issues here (were instead of where, these instead of those)

Line 320: ENHanCEd is capitalised incorrectly.

Line 412: comparing similarity to other COVs in a host to similarity to all viruses is not equivalent to comparing similarity to those from a host and those not in a host, it is equivalent to comparing the background similarity.

Reviewer #1

In this manuscript, the authors use multiple similarity learners to predict the potential pool of mammalian hosts that are capable of supporting coronavirus infection. Using these approaches, the authors show that there are manifold more potential coronavirus-host associations and SARS-CoV-2 recombination hosts than have been currently characterized or described. These findings are of particular value considering the current pandemic.

Overall, the study is elegant, informative, and thorough, and the writing is strong. One comment that may be particularly germane to the current pandemic is that the authors did not include a comparison of ace2 expression vs. similarity to human ACE2 in their algorithms, which would greatly strengthen the argument for a largely uncharacterized pool of potential hosts of SARS-CoV-2. If this is planned for a future study, a discussion of its potential application is warranted.

Our response: We thank the reviewer of their remarks. We agree with their observation that a comparison of ACE2 expression and sequence homology to human ACE2 would have increased the accuracy of our framework. However, there are two fundamental reasons why it was not included in this study (but shall be in ongoing follow-up work).

- 1) Our goal was to include as many mammalian species and coronaviruses as possible in our work to deliver the ‘big picture’ of all susceptible hosts to all coronaviruses and therefore predict recombination hosts for all coronaviruses. ACE2 (and its orthologous and paralogs in different species) has only been sequenced for a small number of mammals, furthermore, expression analysis and annotation of paralogs has been completed for an even smaller number. Indeed, ACE2 itself is not used for cell entry by many coronaviruses. Consequently, the lack of data for the vast majority of hosts, and lack of application to many coronaviruses, would prevent it from being utilized by our framework (or any framework studying the full range of hosts and coronaviruses). It is however useful for a more focused approach (on which we are currently working) on a small number of well-characterised hosts and ACE2-utilising coronaviruses. Discussion of this has been added to the text.
- 2) Secondly, as mentioned in the discussion, our approach was one of ‘no preconceptions’. This is partly due to the lack of data mentioned in the first point, but also because ACE2 mainly deals with a component of cell entry, there are also many other genes involved in cell entry, as well as unknown genes/factors which are required for a host to be viable (e.g. ability to replicate once inside of the cell, ability to recruit other host cellular machinery, not be too pathogenic and kill the cell/ elicit a strong immune response, etc.). These components are governed by complex combinations of unknown factors. Consequently, no preconceptions, i.e. including all available traits from both host and virus, and allowing our pipeline to learn from all available data

(expressed into similarities), was decided upon as a way of including all practical variables and minimising our (and literature-based) preconception bias on the output.

We have modified our discussion to elaborate on this no preconceptions approach and highlight the potential of ACE2 to be utilised in a more focused study. We have also added to the discussion a description of future work which encompasses the suggestion made here including ACE2 receptor information.

Changes to MS - Discussions (lines 287 - 297): Our method does not make assumptions about which parts of the coronavirus or host genomes are important, or integration of receptor (e.g. ACE2) information, or focusing on certain groups of hosts (e.g. bats or primates). This ‘no-preconceptions’ approach enables us to analyse without being restricted by our current incomplete knowledge of the specific biological and molecular mechanisms which govern host-virus permissibility. Current restrictions include lack of sequencing, annotation and expression analysis of receptor (e.g. ACE2) in the vast majority of hosts, uncertainty over the receptor(s) utilised by many coronaviruses, and knowledge of other factors leading to successful replication once the virus has entered the host cell. Whilst some of these details are known for a very limited number of well-studied hosts and coronaviruses, they are not for the vast majority, consequently, a study aiming for breadth of understanding across all mammalian hosts and coronaviruses is unable to utilise these limited data.

Changes to MS - Discussions (lines 327-334): To follow on from this work, we are investigating coronavirus-host interactions in two separate directions. The first is to expand our host range to include avian species, therefore, including the full range of important coronavirus hosts, and to inform our model with a species-level contact network for all hosts (indicating likeliness of a direct interaction). This will give a broader overview of potential coronavirus associations. Secondly, we are focusing our predictions on studying a sub-set of clinically important associations in more depth. This will allow us to utilise more specific information such as receptor and clinical data on the viraemia which are only currently available for well-studied interactions.

Reviewer #3

Wardeh et al provided a methodology for predicting mammalian hosts which are likely to carry coronaviruses. Recombination research is by far the most difficult to model in the field of viral evolution, because it can lead to uncertain judgements on host and phylogeny. I found the study design and the results presented are very novel and interesting. However several comments below should be addressed.

1. Zoonotic coronaviruses do recombine easily. However, please do not use a positive tone to conclude that SARS-CoV-2 is a recombinant virus, as investigations into its origin and natural hosts are clearly under way around the world. It's not easy to conduct this type of study, e.g. it took more than a decade to find out the origin of SARS-CoV.

Our response: We agree with the reviewer and have reduced the strength of the tone in both the abstract and introduction to reflect this uncertainty and ongoing research on the origins of SARS-CoV-2.

Changes to MS - Abstract (lines 14-15): Novel pathogenic coronaviruses – such as SARS-CoV and probably SARS-CoV-2 – arise by homologous recombination between co-infecting viruses in a single cell.

Changes to MS - Introduction (lines 51-52) : Homologous recombination in Spike has been implicated in the generation of SARS-CoV-2¹⁵, although investigations are still ongoing.

2. How do authors define host types? As far as I am aware, coronaviruses could be isolated from non-natural hosts i.e. intermediate hosts, and the intermediate hosts are clearly not responsible for the origin of coronaviruses.

Our response: Due to extreme lack of data on role different host species play in transmission of coronaviruses (and indeed most other viruses), we did not define different types of host. We treated all associations between viruses and hosts in the same manner. Secondly, we don't agree that 'intermediate hosts are clearly not responsible', it seem quite feasible that should an intermediate host have its own natural coronaviruses at a high frequency, and these could readily provide a source of genetic material for recombination when the host is occasionally infected with a different virus. Therefore, we believe there is both a 'data limitation' and biological justification for including all hosts. Text has been added to the discussion to address this.

Changes to MS – Discussion (lines 309-315): 2) Virological knowledge of understudied coronaviruses and their host interactions. For the vast majority of observed virus-host associations it is unknown if these hosts are natural, intermediate, or 'dead-end' hosts. Also unknown are more clinical traits of the infections in the overwhelming majority of associations, such as: pathogenicity, likelihood of infection, virus titre during infection, duration of infection, etc. knowledge of all of these factors

could greatly add to our ability to assess ‘likeliness’ of homologous recombination, however, the available data are too limited for a study with the breadth of interactions we characterise here, and hence were unable to be included.

3. How did the authors select the best algorithm for the prediction?

Our response: Initially we trained a suite of classification algorithms on earlier versions of the data, we selected due to their robustness, scalability, availability, and over-all performance. While no one algorithm emerged as best performing across our initial tests. GBMs performed consistently well. We have since repeated this process with the data used to train and test our final model. Results of these new run produced similar results (attached supplementary figure below). In addition, GBMs are known for their ability to handle non-linearity and high-order interactions between their variables (Allen et al, 2017), and have been used to predict reservoirs of viruses (Babayan et al, 2018), and zoonotic hot-spots (Allen et al, 2017).

References: Babayan, S. A., Orton, R. J. & Streicker, D. G. Predicting reservoir hosts and arthropod vectors from evolutionary signatures in RNA virus genomes. *Science* (80-.). **362**, 577–580 (2018).

Allen, T. *et al.* Global hotspots and correlates of emerging zoonotic diseases. *Nat. Commun.* **8**, 1124 (2017).

Changes to MS – methods (lines 489-491): We selected GBM as our stacking algorithm due to its ability to handle non-linearity and high-order interactions between constituent learners⁴⁸, and have been used to predict reservoirs of viruses⁴², and zoonotic hot-spots⁴⁸.

Changes to ESM – New supplementary figure (S14): we added a new supplementary figure (S14 – attached below) featuring performance metrics of the classification algorithms we included in our selection process.

Figure S14. Comparison of performance metrics of 7 classification algorithms over the held-out test sets ($n = 5$) at 0.5 probability cut-off. We trained 7 classification algorithms: Model Averaged Neural Network (avNNet), Stochastic Gradient Boosting (GBM), Random Forest (RF), Support Vector Machines with radial basis kernel and class weights (SVM-RW), Linear SVM with Class Weights (SVM-LW), SVM with Polynomial Kernel (SVM-P), and Naive Bayes. We selected these algorithms due to their robustness, scalability, availability, and over-all performance. All models were trained and tested via caret R package. While no one algorithm performed best across all five tests, GBM performed well across the 5 tests, thus it was selected to perform the stacking of our similarity-based learners.

4. Recombination could have occurred in any types of hosts and not just limited to a certain type of host. How did the authors define the different types of hosts in their study, i.e. natural hosts, intermediate hosts and dead-end hosts? Are there any literatures showing certain type of hosts are easily targeted by coronaviruses?

Our response: We did not define different types of host. Any association between virus and host is treated the same. Whilst we agree that an understanding of e.g. intermediate vs dead-end could help to inform our model, these data are not available for the overwhelming majority of associations. This level of interaction is only known for the few highly studied associations, so would only be usable in a study focusing on a very small subsection of already well studied data. As this study focused on understanding the breadth of association across all viruses and mammals, it was not possible to be included. We have added a paragraph in the discussion which discusses this and other examples of ‘in-depth’ data which would be useful in determining associations, but do not exist for the majority of species.

Changes to MS – Discussion (lines 309-315): 2) Virological knowledge of understudied coronaviruses and their host interactions. For the vast majority of observed virus-host associations it is unknown if these hosts are natural, intermediate, or ‘dead-end’ hosts. Also unknown are more clinical traits of the infections in the overwhelming majority of associations, such as: pathogenicity, likelihood of infection, virus titre during infection, duration of infection, etc. knowledge of all of these factors could greatly add to our ability to assess ‘likeliness’ of homologous recombination, however, the available data are too limited for a study with the breadth of interactions we characterise here, and hence were unable to be included.

4. Given the receptors for most types of coronaviruses are unknown, it is a bit risky to ‘predict’ different coronaviruses shared by same/similar hosts. Some ‘hosts’ could be only transient hosts that hardly provide any evolution forces into the evolution of coronaviruses. Could the authors give a successful predicted case here?

Our response: As discussed in our response to the previous question, we do not (and cannot) define different types of hosts for the overwhelming majority of associations (and therefore cannot for predictions). Hence, all known hosts are considered ‘observed hosts’, and predicted associations are all considered ‘predicted hosts’, with no further distinction possible. Consequently, this question does not really apply, as demonstrated by the following example: Should parental viruses A and B recombine to form progeny virus C, then the host in which they recombined, if known, would already be an ‘observed host’ for A, B and C. Consequently, we cannot compare any prediction we have made on A, B or C with their host to the known recombination event as the host would all already be included in the ‘observed host’ data.

5. Why there are no birds which are natural hosts for some gamma-coronaviruses?

Our response: We agree with the reviewer about birds being natural hosts for gamma-CoVs. However, we could not produce predictions for avian species in this study due the available data being incomplete, disparate, and requiring excessive amount of time to manually curate sufficient data describing observed and potential avian hosts to the same standard as mammalian data. In its current form, the available data would have resulted in avian models which were incomparable to our mammalian models, therefore the two sets could not have been unified to produce final predictions. We are planning to include all potential mammalian and avian species in follow-on work once this curation has been completed.

Changes to MS - Discussions (lines 327-334): To follow on from this work, we are investigating coronavirus-host interactions in two separate directions. The first is to expand our host range to include avian species, therefore, including the full range of important coronavirus hosts, and to inform our model with a species-level contact network for all hosts (indicating likelihood of a direct interaction). This will give a broader overview of potential coronavirus associations. Secondly, we are focusing our predictions on studying a sub-set of clinically important associations in more depth. This will allow us to utilise more specific information such as receptor and clinical data on the viraemia which are only currently available for well-studied interactions.

6. Could the authors use different colors in Figure 3? It is a bit hard to see any pattern in this figure.

Our response: We have adjusted the colour scheme for Figure 3. Please note, that in order to be consistent with the remainder of our analyses, we have adjusted figure 3 – panel B to reflect predicted network at probability ≥ 0.90 , so that reported probabilities match the remainder of the text and supplementary materials.

Changes to MS – Updated Figure 3. Attached below.

Figure 3 – Bipartite networks linking coronaviruses with mammalian hosts. Panel A: Original bipartite network based on known/observed virus-host associations extracted from meta-data accompanying genomic sequences and supplemented with publications data from the ENHanCED Infectious Diseases database (EID2). Panels B, C and D show predicted bipartite networks using our predicted virus-host associations at different cut-offs: 0.90, 0.75 and 0.5, respectively, for mean probability of associations.

7. What are un-classified coronaviruses?

Our response: We adopted NCBI taxonomy of viruses, which is the same one used in EID2. Unclassified coronaviruses here refer to coronaviruses which are positioned in the NCBI taxonomy under the node: unclassified Coronavirinae (no rank/artificial node, equivalent name: unclassified coronaviruses). At times of analyses, we were able to capture six viruses in this category for which full genome sequences were available at the time: bat coronavirus, hypsugo bat coronavirus hku25,

bat coronavirus bm48-31/bgr/2008, swine acute diarrhea syndrome coronavirus, bat coronavirus ratg13, and swine acute diarrhea syndrome related coronavirus.

Changes to MS – Methods (lines 356-357): Of our included species, six in total were unclassified Coronavirinae (unclassified coronaviruses).

Changes to Figures – We adjusted figure 3 legend (attached above), and the caption of figure 2.

Reviewer #4

In this paper, the authors use data on 411 coronavirus strains, 876 hosts and network of known host-pathogen interactions between these entities to predict host-pathogen interactions that may exist but have not yet been documented. I have a few comments on the study design and some minor corrections.

Firstly, I appreciate that the motivation of the study was to identify different coronaviruses that have the potential to recombine within the same host and thus gain the ability to infect humans, but it seems a shame to have not included humans in the host-pathogen network. Surely the potential for coronaviruses to coinfect and recombine within humans would also be relevant here, and understanding how we compare to these other hosts in terms of the diversity of coronaviruses that are predicted to infect us would be interesting. It may also give a better opportunity to test whether host range predictions are correct, assuming the diversity of viruses that infect humans is better studied than that of animals.

Our response: Our apologies for not being clear in the manuscript, we did indeed include humans (and lab-rodents) in all stages of the analysis, and hence the results from these predictions can be used to test predictions made as the review suggests. The confusion has arisen because we excluded humans from figures 1 and 2 presented in the main manuscript. The paper was intended to focus on recombination potential in non-human hosts. Therefore, we excluded humans (and lab-rodents) from these figures so to enhance the visualisation of these potential recombination hosts. We have replicated the figure to include humans and lab rodents in the supplementary materials (Supplementary Figure S13 – attached below). We have re-emphasised that these data were included in the analysis in the figure legend. We have also included a supplementary dataset (Supplementary Data-S5) containing all predicted coronavirus associations with humans and their probabilities (starting from 0.5 threshold) for the suggested future testing opportunity.

Changes to ESM – New Supplementary Figure S14. Attached below.

Figure S14. Model predictions for potential hosts of SARS-Cov-2 – including humans and lab rodents. Predicted hosts are grouped by order (inner circle). Middle circle presents probability of association between host and SARS-CoV-2 (Grey scale indicates predicted associations with probability in range $>0.5 - \leq 0.75$. Red scale indicates predicted associations with probability in range $>0.75 - <0.9$. Blue to purple scale present indicates associations with probability ≥ 0.9). Yellow bars represent number of coronaviruses (species or strains) observed to be found in each host. Blue stacked bars represent other coronaviruses predicted to be found in each host by our model. Predicted coronaviruses per host are grouped by prediction probability into three categories (from inside to outside): ≥ 0.9 , $>0.75 - <0.9$, and $>0.5 - \leq 0.75$.

Our response 2: In addition to this addressing of the reviewer’s comment as per the above section, we have also added an additional section to the discussion and a supplementary dataset (Supplementary Data-S6) which we feel provides a better opportunity for future testing of our predictions. Whilst this manuscript was under review another group has bioinformatically demonstrated the high ability for SARS-CoV-2 to homologously recombine with MERS-CoV, which could potentially generate a highly pathogenic virus with the human-human transmission of SARS-CoV-2 and the high patient-mortality phenotype of MERS-CoV. This is an important finding and our studies synergistically combine to identify ‘what to look for’ and ‘in which species to look’. We feel

that this is a highly important area for surveillance/study going forward and consequently provides an ideal opportunity for testing our predictions.

Changes to MS – Discussion (lines 256-265): As an example of the utilisation of our model from the perspective of likely future viral homologous recombination events, Banerjee et al³⁹ bioinformatically identified potential genomic regions of homologous recombination between MERS-CoV and SARS-CoV-2. They highlighted a significant risk of the highly human-to-human transmissible SARS-CoV-2 acquiring the considerably more pathogenic (i.e. in terms of case-fatality rate) phenotypes of MERS-CoV. The work presented here identifies 102 [75; 44] potential recombination hosts (excluding humans and laboratory rodents) of the two viruses. Together, our work and Banerjee et al³⁹, provide evidence for both the production of a potentially severe future recombinant coronavirus, and identify the hosts in which this threat is most likely to be generated (see supplementary dataset 6). We recommend monitoring for this event.

New reference - 39: Banerjee, A. et al. Predicting the recombination potential of severe acute respiratory syndrome coronavirus 2 and Middle East respiratory syndrome coronavirus. *J. Gen. Virol.* jgv001491 (2020). doi:10.1099/jgv.0.001491.

Secondly, the predicted expansion to the connectivity of the host-pathogen network is substantial. I would prefer the analysis to focus more on the predictions made at the 0.9 probability level than the 0.5, to add more confidence to the conclusions from the modelling side.

Our response: We agree that predictions made at 0.9 are more reliable, with regards to predicted expansion, than those at 0.5. However, we decided to include both at all times in our reporting (e.g. throughout the discussion and in figures 1, 3, table 1, and in the relevant supplementary materials). This is because only looking at 0.9 would miss a lot of real associations; an association at 0.5 for an important coronavirus is certainly ‘worth investigating’ for example. In addition to these two, we also now add in 0.75, which serves as a balance between them. Providing all three gives the reader the ability to determine what is appropriate for their application. Alongside this change, we have altered figure 2 to demarcate these three ‘cut-offs’ (0.5-0.75, 0.75-0.9, and 0.9-1), to be consistent with the rest of the updated manuscript and highlight the probability cut-offs. Hence, the figure has been re-coloured accordingly. Figure 1 has also been minorly altered to be consistent with this change. Table 1, and figure 3 were updated with network and metrics generated at 0.9 cut-offs replacing these generated at 0.95 cut-off to maintain consistency.

In addition to figure changes, we updated results and discussion so that predictions at the three probability cut-offs: >0.5 , >0.75 and ≥ 0.9 are listed for all prediction made. Furthermore, we updated supplementary datasets S2-S4 to incorporate these cut-offs.

Changes to MS – New Figure 2. Attached below.

Figure 2 – Observed and predicted mammalian hosts for coronaviruses. Columns present mammalian hosts in four categories: Artiodactyla & Perissodactyla (top 10 hosts by number of predicted coronaviruses that could be found in each host); Carnivora (top 15 hosts), Chiroptera (top 15 hosts, each predicted to host 50 or more coronavirus species or strain), and others (top 5). Rows present viruses ordered into five taxonomic groups: Alphacoronaviruses, Betacoronaviruses, Deltacoronaviruses, Gammacoronaviruses and unclassified Coronavirinae. Yellow cells represent observed associations between the host and the coronavirus. Grey/red/blue cells indicate the probability of predicted associations in three increasing probability ranges. White cells indicate no known or predicted association between host and virus (beneath cut-off probability of 0.5). These results exclude humans and lab rodents. Supplementary figure S15 illustrates full results including these hosts.

New Table – Table 1. Attached below.

Table 1 – Observed and predicted number of hosts of SARS-CoV-2 (by mammalian order), and observed and predicted number of hosts with 10 or more coronaviruses (from our set of 411 species or strains). Numbers are presented at three probability cut-offs: >0.5, >0.75 & ≥0.9. Values in brackets are generated by subtracting/adding standard deviation (SD) from the mean probability of the ensemble (100 runs) and generating predictions at the listed cut-offs.

Mammalian order	Observed & predicted hosts of SARS-CoV-2			Observed & predicted hosts with 10 or more coronaviruses		
	cut-off >0.5	cut-off >0.75	cut-off ≥0.9	cut-off >0.5	cut-off >0.75	cut-off ≥0.9
Artiodactyla	18 (-8/+0)	15 (-11/+3)	8 (-8/+10)	20(-2/+0)	20(-13/+0)	16(-15/+4)

Carnivora	37 (0/+0)	37 (-14/+0)	37 (-30/+0)	35(-22/+2)	20(-14/+17)	10(-9/+25)
Chiroptera	25 (-19/+0)	13 (-12/+12)	3 (-3/+22)	129(-87/+53)	56(-38/+105)	30(-26/+105)
Eulipotyphla	5 (-4/+0)	3 (-3/+2)	1 (-1/+4)	5(0/+0)	5(-4/+0)	3(-3/+2)
Lagomorpha	2 (-1/+0)	2 (-2/+0)	1 (-1/+1)	2(0/+0)	2(-1/+0)	1(-1/+1)
Perissodactyla	2 (0/+0)	2 (-2/+0)	2 (-2/+0)	2(0/+0)	2(-2/+0)	2(-2/+0)
Pholidota	1 (0/+0)	1 (-1/+0)	0 (0/+1)	1(0/+0)	1(-1/+0)	0(0/+1)
Primates (non-human)	4 (0/+0)	4 (-1/+0)	4 (-4/+0)	4(0/+0)	4(-4/+0)	4(-4/+0)
Rodentia (excluding laboratory species)	32 (-9/+0)	26 (-17/+6)	16 (-14/+16)	33(-4/+3)	30(-27/+6)	19(-19/+15)

Changes to MS – Results. We incorporated the three cut-offs throughout – we list one example **lines (102-110)**: Overall, our model predicted 4,438 (SD=-1,903/+2,256, cut-off>0.5) previously unobserved associations that potentially exist between 300 (SD=0/+3) mammals and 204 coronaviruses (species or strains, SD=-60/+13). The number of unobserved associations at probability cut-offs >0.75, and ≥ 0.9 were: 3,087 (-1747/+2391) between 300 (-16/+0) mammals and 181 (-127/+26) coronaviruses, and 1,989 (-1692/+2784) between 299 (-128/+1) mammals and 92 (-66/+114) coronaviruses, respectively. Our model predicts there are 115 (0/+3) [115 (-4/+0), 115 (-29/+0), at cut-offs >0.75, and ≥ 0.9] mammalian species with no previously observed associations with the 411 input viruses (here after we display results derived from >0.5 cut-off; results obtained at >0.75, and ≥ 0.9 cut-offs are presented in square brackets).

Changes to Results – Validation (lines 164-168): We validated our analytical pipeline externally against 20 held-out test-sets (as described in method section below). On average, our GBM ensemble achieved AUC = 0.948 (± 0.029 SD), 0.944 (± 0.024), 0.922 (± 0.024); TSS=0.896 (± 0.057), 0.887 (± 0.048), 0.844 (± 0.048); and F-Score = 0.102 (± 0.049), 0.141 (± 0.055), 0.191 (± 0.054), at probability cut-offs >0.5, >0.75 and ≥ 0.9 , respectively (supplementary figures S7 – S12).

Changes to MS – Discussion. We incorporated the three cut-offs throughout – we list one example **lines (172-175)**: We predict 11.54-fold increase – prediction cut-off >0.5 [8.33-fold; 5.72-fold, cut-offs >0.75, ≥ 0.9 , respectively, cut-offs presented in this format hereafter], leading to the prediction that there are many more mammalian species than are currently known in which more than one coronavirus can occur.

In the Supplementary data, tests on held-out datasets 1 and 3 yield very different performance metrics than tests 2, 4 and 5 - can the authors comment on why this is?

Our response: This is due to the stochastic effect of small number of tests. We have repeated our tests 15 more times (= 20 in total) drawing in each 15% of known coronavirus-mammalian associations (at random). We attach the corresponding supplementary figure (now supplementary figure S7) below. In addition, to add to our response to other comments we added two additional supplementary figures (S8 and S9) illustrating performance metrics of our meta-ensembles at 0.75 and

0.9 cut-offs, respectively. We have also changed the y-axis scales to begin from ‘0’, the previous scale made the differences appear more exaggerated than they truly are.

Changes to ESM – New supplementary figures (S7-S9). Attached below.

Figure S7. Performance assessment over the held-out test sets (n =20) at 0.5 probability cut-off. Violin plots and points are coloured by test number. In each test we created a held-out test comprising 15% of all data (including 15% of all observed associations, and 15% of all unknown associations). Our learners were trained with the remainder 85% of the observed (and unknown) associations, and our meta-ensemble was then trained with predictions of these learners (10-fold cross validation, 100 repeats). Performance metrics were then computed against the held-out test set and reported here.

Figure S8. Performance assessment over the held-out test sets (n =20) at 0.75 probability cut-off. Violin plots and points are coloured by test number. In each test we created a held-out test comprising 15% of all data (including 15% of all observed associations, and 15% of all unknown

associations). Our learners were trained with the remainder 85% of the observed (and unknown) associations, and our meta-ensemble was then trained with predictions of these learners (10-fold cross validation, 100 repeats). Performance metrics were then computed against the held-out test set and reported here.

Figure S9. Performance assessment over the held-out test sets (n =20) at 0.9 probability cut-off. Violin plots and points are coloured by test number. In each test we created a held-out test comprising 15% of all data (including 15% of all observed associations, and 15% of all unknown associations). Our learners were trained with the remainder 85% of the observed (and unknown) associations, and our meta-ensemble was then trained with predictions of these learners (10-fold cross validation, 100 repeats). Performance metrics were then computed against the held-out test set and reported here.

I would like to see confusion matrices as well as the metrics presented in Figure S6, to get a better feel for the raw numbers of true/false positives/negatives.

Our response. We produced confusion matrices for our original and new sets of tests (20 in total) at three probability cut-offs: 0.5, 0.75 and 0.9. These matrices were generated taking the mean probability of the meta-ensembles (across 100 runs), per each test. We attach these supplementary figures in our changes below.

Changes to ESM – New supplementary figures (S10-S12). Attached below.

Figure S10. Confusion matrices produced over the held-out test sets (n =20) at 0.5 probability cut-off. Confusion matrices were generated by taking the mean probability (across 100 runs) of our GBM meta-ensemble, predicted values >0.5 where considered positive (Yes), and those <=0.5 were considered negative (No). Colours in the above matrices indicate agreement between the predicted and the known associations (blue), or no agreement (blue). Transparency (alpha) indicates probability of agreement (the more times the two sets agreed, in relation to the total space (of yes or no) the more opaque the matrix cell).

Figure S11. Confusion matrices produced over the held-out test sets (n =20) at 0.75 probability cut-off. Confusion matrices were generated by taking the mean probability (across 100 runs) of our GBM meta-ensemble, predicted values>0.75 where considered positive (Yes), and those <=0.75 were considered negative (No). Colours in the above matrices indicate agreement between the predicted and the known associations (blue), or no agreement (blue). Transparency (alpha) indicates probability of agreement (the more times the two sets agreed, in relation to the total space (of yes or no) the more opaque the matrix cell.

Figure S13. Confusion matrices produced over the held-out test sets (n =20) at 0.9 probability cut-off. Confusion matrices were generated by taking the mean probability (across 100 runs) of our GBM meta-ensemble, predicted values ≥ 0.9 where considered positive (Yes), and those < 0.9 were considered negative (No). Colours in the above matrices indicate agreement between the predicted and the known associations (blue), or no agreement (blue). Transparency (alpha) indicates probability of agreement (the more times the two sets agreed, in relation to the total space (of yes or no) the more opaque the matrix cell).

The ability to infect a host is one factor, but the amount of time a given virus spends in a given host, and the viral load in that host, should also matter in terms of the probability to two strains coinhabiting a host and recombining.

Our response: Whilst we agree that an understanding of e.g. viremia length and viral load could help to inform our model in terms of likelihood of viruses coming into contact, these data are not available for the overwhelming majority of viruses and their associations. It is only known for the few highly studied associations, so would only be usable in a study focusing on a very small subsection of already well studied data. As this study focused on understanding the breadth of association across all viruses and mammals, it was not possible to include these data. We have added two paragraphs in the discussion which discusses this and other examples of ‘in-depth’ data which would be useful in determining associations, but do not exist for the majority of species, so is not possible to include in a broad study, but will be included in future more focused work.

Changes to MS – Discussion (lines 309-315): 2) Virological knowledge of understudied coronaviruses and their host interactions. For the vast majority of observed virus-host associations it is unknown if these hosts are natural, intermediate, or ‘dead-end’ hosts. Also unknown are more clinical traits of the infections in the overwhelming majority of associations, such as: pathogenicity, likelihood of infection, virus titre during infection, duration of infection, etc. knowledge of all of these factors could greatly add to our ability to assess ‘likeliness’ of homologous recombination, however, the available data are too limited for a study with the breadth of interactions we characterise here, and hence were unable to be included.

Changes to MS - Discussions (lines 327-334): To follow on from this work, we are investigating coronavirus-host interactions in two separate directions. The first is to expand our host range to include avian species, therefore, including the full range of important coronavirus hosts, and to inform our model with a species-level contact network for all hosts (indicating likeliness of a direct interaction). This will give a broader overview of potential coronavirus associations. Secondly, we are focusing our predictions on studying a sub-set of clinically important associations in more depth. This will allow us to utilise more specific information such as receptor and clinical data on the viraemia which are only currently available for well-studied interactions.

I would be interested to see the probabilities assigned here compared to any available data on the efficiency of CoVs to infect different hosts - are they correlated?

Our response: We are unsure of exactly what the reviewer is asking here, so have attempted to answer both possibilities.

1) If the reviewer is referring to comparing the total number of hosts a virus could have against predictions: We agree that comparing efficiency of CoVs to infect different numbers/diversity of hosts would aid in our prediction pipeline. However, this is not feasible to do beyond taking into account the known hosts. This is because no CoV has been tested for its ability to infect even a relatively small number of its potential hosts; we are working with a very incomplete dataset for all viruses and all hosts. Therefore, for any given virus we actually do not know if it is unable to infect more hosts, or it just has not been observed or looked for; hence cannot compare it to any predictions on breadth of associations. In addition, the network components of our models (as described in methods and supplementary note 2), incorporate all available data on which CoVs infect which hosts, and how many known hosts (as part of the network topology expressed by the resulting embeddings of both hosts and CoVs nodes). Moreover, phylogenetic and ecological diversity of relatedness of our hosts were also accounted for, to an extent, by the inclusion of host phylogenetic, and ecological traits in our similarity learners (from the mammalian point of view).). We believe that this is the greatest extent to which we can account for the incomplete data.

2) Alternatively, if the reviewer is referring to how well our framework predicts already known hosts which have been observed: we have added a new supplementary figure (S16, attached below) showing the full range of predicted associations, and highlighting the probability of the observed associations we attach this figure below. The percent of observed associations (known to exist between the focal mammal and the focal coronavirus) were predicted by the final model (trained with all available data) were as follows: 95.25% were predicted with probability ≥ 0.9 , 97.37% were predicted with probability > 0.75 , and again 97.37% were predicted with probability > 0.5

Changes to ESM – new Supplementary Figure S16. Attached below.

Figure S16. Final model predictions and observed coronavirus-mammalian associations. Yellow circles present observed associations (known) between coronaviruses and mammalian hosts. Grey circles indicate associations predicted by the final model with probability > 0.5 and < 0.7 . Red circles indicate associations predicted with probability < 0.7 and < 0.9 . Blue-purple circles indicate associations predicted with probability ≥ 0.9 . Y-axis represent the probability produced by the final model (trained with all available data, with 10-fold cross validation) – ranging from 0 to 1. X-axis represent the order of the mammalian host, as follows (clockwise): Artiodactyla, Carnivora, Chiroptera, Eulipotyphla, Lagomorpha, Perissodactyla, Pholidota, Primates, and Rodentia. Order

silhouettes were obtained from *phylopic*: <http://phylopic.org/>. The percent of observed associations (known to exist between the focal mammal and the focal coronavirus) were predicted by the final model (trained with all available data) were as follows: 95.25% were predicted with probability cut-off ≥ 0.9 , whereas 97.37% were predicted with cut-offs: >0.75 , and >0.5 .

Did the individual components in the meta-ensemble agree or disagree about host-pathogen links? It wasn't clear whether multiple independent lines of evidence led to these conclusions, or whether a single source of support tended to be sufficient to draw a link.

Our response: Our meta-ensemble “*blended*” the confidence estimated by its constituent components (similarity learners) in non-linear manner (via the GBM algorithm) in each of its runs (100 in total, predictions based on the mean probability of these runs). In order to illustrate how these components influenced the meta-ensemble, we generated an additional supplementary figure (Figure S6, attached below) to complement Figure S5 (which indicated the relative influence each meta-learner had on the meta-ensemble). This new supplementary figure illustrates the partial dependence of our ensemble on each of its individual components. Partial dependence measures the response for an individual variable in a machine-learning model (here GBM), while holding all other variable constant. Partial dependence plots visualise the non-linear relationships between each similarity-learner in our meta-ensemble and the response variable (whether a given coronavirus could potentially be found in a given mammalian host).

Changes to ESM – new Supplementary Figure (Figure S6). Attached below.

Figure S6. Partial dependence plots showing the influence on coronavirus-mammal associations for all similarity learners in all runs of our meta-ensemble (GBM). X axes show the range of values of our similarity learners (0 to 1). Y axes show the effect on the probability of coronavirus-mammal association (0 to 1) from that learner. Individual lines show the partial dependence per each run of the ensemble. The smoothed line (smoothed conditional means) indicates the overall trend of partial dependence between our response variable and the learner. Partial dependence measures the response for an individual variable in a machine-learning model (here GBM), while holding all other variable constant. Partial dependence plots visualise the non-linear relationships between each similarity-learner in our meta-ensemble and the response variable (whether a given coronavirus could potentially be found in a given mammalian host).

Minor corrections:

Figure 1: the colouring of the number of coronaviruses predicted to be in each host may be a bit off here, as it looks like there are no predicted links with ≥ 0.9 support.

SARS-CoV-2 is at one point written as SARS-COV2 - please make these consistent

Corrected to SARS-CoV-2

Line 312 - some spelling issues here (were instead of where, these instead of those)

Paragraph corrected

Line 320: ENHanCED is capitalised incorrectly.

Corrected throughout text.

Line 412: comparing similarity to other COVs in a host to similarity to all viruses is not equivalent to comparing similarity to those from a host and those not in a host, it is equivalent to comparing the background similarity.

Corrected.

Reviewers' Comments:

Reviewer #3:

Remarks to the Author:

The authors have addressed all my concerns. Congratulations.

Reviewer #4:

Remarks to the Author:

I appreciate the level of work that has gone into this revised manuscript and feel that the additional analyses add great clarity to the manuscript. I am still not wholly convinced by the level of confidence in the findings, however.

Methods (lines 489-491) - many machine learning methods are known to handle non-linear and high-order interactions well, so this isn't a good justification for the specific choice of method.

The lack of a clear ground truth here makes assessment of this work very challenging. At the 0.5 probability level (assuming the probabilities have been correctly calibrated, which is unlikely, especially with strongly imbalance classes), we would hope to see roughly half of predicted interactions validated as real, but looking at the confusion matrices in Fig S10-13, a much smaller proportion of interactions have actually been observed. This is both the major finding of the paper, and a typical indicator that the algorithm is "inaccurate". How can the reader tell the difference here between a new discovery and an algorithm with a high false-positive rate?

Fig S16 - the previously observed host-pathogen interactions mostly seem to have scored with probabilities well over 0.9. This suggests to me that interactions should be reported at the $P > 0.98$ level or similar, to better match the probability distribution assigned to true-positives.

"This 'no-preconceptions' approach enables us to analyse without being restricted by our current incomplete knowledge of the specific biological and molecular mechanisms which govern host- virus permissibility" - This is presented as a strength, but it's also a big limitation. These predictions are being made with limited information, so the authors need to be careful with the level of confidence they ascribe to their findings. The findings don't show that the number of interactions and potential recombination hosts are underestimated. They predict this, but these predictions need to be validated before we can say with confidence that host interactions were previously underestimated.

Discussion - "potential hosts" should be replaced or qualified with "predicted hosts", as potential could refer to a known host that could potentially be infected and facilitate recombination, rather than a predicted host that has no empirical evidence to support it yet. E.g. "indicating that observed data are missing 31.5-fold of the total number of potential recombination hosts" should be changed to "indicating that observed data are predicted to be missing 31.5-fold of the total number of potential recombination hosts"

"Furthermore, our results show that the number of viruses which could potentially recombine even within these known hosts has been significantly under-ascertained, indicating that there still remains significant potential for further novel coronavirus generation in future from current known recombination hosts." versus

"Furthermore, our results suggest that the number of viruses which could potentially recombine even within these known hosts may have been significantly under-ascertained, which would mean that there still remains significant potential for further novel coronavirus generation in future from current known recombination hosts."

Reviewer #4

I appreciate the level of work that has gone into this revised manuscript and feel that the additional analyses add great clarity to the manuscript. I am still not wholly convinced by the level of confidence in the findings, however.

Methods (lines 489-491) - many machine learning methods are known to handle non-linear and high-order interactions well, so this isn't a good justification for the specific choice of method.

Our response: We thank the reviewer for their comments. We have provided an answer to this point in our previous response to reviewer #3. We modified related section in methods for clarity.

We initially trained a number of algorithms on earlier versions of the data. We selected a range of 7 algorithms due to their robustness, scalability, availability, and over-all performance. While no one single algorithm emerged as best performing across our initial tests, GBMs performed consistently well. We then repeated this process with the data used to train and test our final model (same splits as the initial 5 tests we have performed). Results of these new runs produced similar results (supplementary figure S14).

Figure S14. Comparison of performance metrics of 7 classification algorithms over the held-out test sets (n = 5) at 0.5 probability cut-off. We trained 7 classification algorithms: Model Averaged Neural Network (avNNet), Stochastic Gradient Boosting (GBM), Random Forest (RF), Support Vector Machines with radial basis kernel and class weights (SVM-RW), Linear SVM with Class Weights (SVM-LW), SVM with Polynomial Kernel (SVM-P), and Naive Bayes. We selected these algorithms due to their robustness, scalability, availability, and over-all performance. All models were trained and tested via caret R package. While no one algorithm performed best across all five tests, GBM performed well across the 5 tests, thus it was selected to perform the stacking of our similarity-based learners.

Changes to MS – methods (lines 502-506): We selected GBM as our stacking algorithm following an assessment of seven machine-learning algorithms using held-out test-sets (20% of known associations randomly selected, n=5 - supplementary figure S14). In addition, GBM is known for its ability to handle non-linearity and high-order interactions between constituent learners⁵², and have been used to predict reservoirs of viruses⁴⁶, and zoonotic hot-spots⁵².

The lack of a clear ground truth here makes assessment of this work very challenging. At the 0.5 probability level (assuming the probabilities have been correctly calibrated, which is unlikely, especially with strongly imbalance classes), we would hope to see roughly half of predicted interactions validated as real, but looking at the confusion matrices in Fig S10-13, a much smaller proportion of interactions have actually been observed. This is both the major finding of the paper, and a typical indicator that the algorithm is “inaccurate”. How can the reader tell the difference here between a new discovery and an algorithm with a high false-positive rate?

Our response: We agree with the reviewer with regards to the importance of ground truthing – but this means comparison of our predictions with the “truth”, yet this is not known. We believe that our dataset is the best available globally, and indeed other researchers have used it to ground-truth their own predictions (e.g., Gibb et al, 2020, and Albery et al ,2020). In the absence of a better dataset than our own against which we can measure our predictions, we have used a different modelling approach which takes as “true” an observed association, and takes as “unknown” (as opposed to “untrue”) an unobserved association. Hence, we have focused mostly on validation against held-out test-sets. This relaxed our approach when predicting these unknowns (e.g., we did not penalise when training if our methods predicted as “true” an unknown, as would have been the case if training/ground -truthing with a gold-standard/complete dataset).

Probability cut-offs: The choice of these cut-offs was so that we provide a range of options, from 0.5 to 0.9 (adjusted to 0.9821, explained below). For clarity, a cut-off probability of 0.5 means that where the meta-ensemble gives the association a >0.5 probability of being positive, we considered the association to be positive (1); and where it is <0.5, we considered it a negative (0). Taking those with model probability>0.5 which we predict to be true, we would logically expect more than half to be true (if our model works, and ideally significantly more than half – as highlighted by our testing process), but this ‘truth’ will in many cases remain unobserved. It is not the case that we expect half or more to **have been observed**, as stated by the referee. These cut-offs and probabilities do not correspond to the known or the true (cannot be known at the moment) distribution of coronaviruses-mammalian hosts associations. Please also note we did not train our ensemble with the full datasets (training/testing), but rather we bootstrapped the process (100 replicate models with balanced samples). Furthermore, we are providing our full set of predictions with their raw probabilities as part of our data/code release, so that they can be utilised to fit different purposes.

Readers clarity: Without a gold-standard, including sufficient number of true negatives (not just known associations), it is not possible to know if our predictions are true or false-positives. Testing is therefore required to differentiate the two scenarios. We are however, reassured that as testing takes place a number of our predictions are being validated (as highlighted in our changes below).

References:

Gibb, R. et al. Zoonotic host diversity increases in human-dominated ecosystems. *Nature* (2020). doi:10.1038/s41586-020-2562-8

Albery, G. F., Eskew, E. A., Ross, N. & Olival, K. J. Predicting the global mammalian viral sharing network using phylogeography. *Nat. Commun.* 11, 1–9 (2020).

Changes - Discussion (lines 333-339): Recent testing of potential mammalian hosts for their susceptibility to SARS-CoV-2 has confirmed a number of our predictions, for example: *Nyctereutes procyonoides*^{41,42}; Bovines (e.g. *Bison bonasus*, *Bos taurus*, *Bos indicus*, *Bubalus bubalis*), *Capra hircus*, *Equus caballus*, *Lama (Vicugna) pacos*, *Manis javanica*, *Oryctolagus cuniculus*, *Panthera leo*, *Rousettus leschenaultii*, *Sus scrofa*, and *Vulpes vulpes*⁴²; *Chlorocebus aethiops*, *Neovison vison*, *Macaca mulatta*

and *Rousettus aegyptiacus*⁴³. While limited in number, these post hoc confirmations adds confidence to our framework and its predictions. As more host screening is performed in future, it will provide further validation of our predictions.

Fig S16 - the previously observed host-pathogen interactions mostly seem to have scored with probabilities well over 0.9. This suggests to me that interactions should be reported at the $P > 0.98$ level or similar, to better match the probability distribution assigned to true-positives.

Our response: We thank the reviewer for their suggestions. To make our methods applicable to changing data, we chose the probability cut-off that would capture 90% of known positive associations (cut-off ≥ 0.9812). We updated relative sections in MS, ESM, and supplementary datasets to reflect this change.

When looking at our cut-offs as a range: 0.5 maximises our model's ability to detect known positives, as evident from the detailed testing we have undertaken (reported in MS, detailed in ESM, and summarised below). Whereas the higher end (0.9812) focuses on enhancing F-score (balancing number of predicted knowns with predicted unknowns).

To illustrate - when applying these cut-offs in our 20 tests performed with 20% held-out test-sets, the range of our performance metrics was as follows:

Cut-off	metric					
	AUC	F-Score	TSS	Sensitivity/ recall	Specificity	Precision
>0.5	0.948 \pm0.029	0.102 \pm 0.049	0.896 \pm0.057	0.935 \pm0.038	0.962 \pm0.044	0.054 \pm 0.027
>0.75	0.944 \pm 0.024	0.141 \pm 0.055	0.887 \pm 0.048	0.906 \pm 0.045	0.982 \pm 0.014	0.077 \pm 0.032
≥ 0.9821	0.843 \pm 0.045	0.283 \pm0.061	0.687 \pm 0.091	0.691 \pm 0.09	0.996 \pm 0.001	0.18 \pm0.044

Therefore, we believe the range of cut-off presented provide the reader with sufficient data to prioritise either detecting all known positives or minimising the number of predicted unknowns. As the choice of probability cut-offs is often made to compromise between these aspects of performance, we also presented predictions made with a cut-off > 0.75 .

Changes – Abstract (line 19-22): We predict that there are 11.5-fold more coronavirus-host associations, over 30-fold more potential SARS-CoV-2 recombination hosts, and over 40-fold more host species with four or more different subgenera of coronaviruses than have been observed to date at > 0.5 mean probability cut-off (2.4-, 4.25- and 9-fold at > 0.9821).

Changes - Methods (lines 531-534): We selected three probability cut-offs for our meta-ensemble: 0.50, 0.75, and 0.9821. One extreme of our cut-off range (0.5) maximises the ability of our ensemble to detect known associations (higher AUC, lower F-score). The other (0.9821) is calculated so that 90% of known positives are captured by our ensemble while reducing the number of predicted associations that are unknown (higher F-Score, lower AUC).

Changes - Results: We replaced all results reported with 0.9 probability cut-off, with new results at 0.9821 cut-off. **Example (lines 102-110):** Overall, our pipeline predicted 4,438 (SD=-1,903/+2,256, cut-off > 0.5) previously unobserved associations that potentially exist between 300 (SD=0/+3) mammals and 204 coronaviruses (species or strains, SD=-60/+13). The number of unobserved associations at probability cut-offs > 0.75 , and ≥ 0.9821 were: 3,087 (-1747/+2391) between 300 (-16/+0) mammals and 181 (-127/+26) coronaviruses, and 601 (-412/+3723) between 224 (-91/+76) mammals and 31 (-7/+171) coronaviruses, respectively. Our model predicts there are 115 (0/+3) [115 (-4/+0), 96 (-31/+19), at cut-offs > 0.75 , and ≥ 0.9821] mammalian species with no previously observed associations with the 411 input viruses (here after we display results derived from > 0.5 cut-off; results obtained at > 0.75 , and ≥ 0.9821 cut-offs are presented in square brackets).

Changes - Discussion: We replaced all results reported with 0.9 probability cut-off, with new results at 0.9821 cut-off. **Example (lines 173-174):** We predict 11.54-fold increase – prediction cut-off > 0.5 [8.33-fold; 2.43-fold, cut-offs > 0.75 , ≥ 0.9821 , respectively, cut-offs presented in this format hereafter].

Changes – Figures: We changed all MS figures so that the higher end in each figure is derived from the new cut-off, replacing 0.9.

Changes – ESM & supplementary data. We updated supplementary figures corresponding to MS figures (S14, S15), and metrics figures (S9, S12, attached below). We also updated supplementary datasets where we have included probability cut-offs.

Figure S9. Performance assessment over the held-out test sets (n =20) at 0.9821 probability cut-off. Violin plots and points are coloured by test number. In each test we created a held-out test comprising 15% of all data (including 15% of all observed associations, and 15% of all unknown associations). Our learners were trained with the remainder 85% of the observed (and unknown) associations, and our meta-ensemble was then trained with predictions of these learners (10-fold cross validation, 100 repeats). Performance metrics were then computed against the held-out test set and reported here.

Figure S12. Confusion matrices produced over the held-out test sets (n =20) at 0.9821 probability cut-off. Confusion matrices were generated by taking the mean probability (across 100 runs) of our GBM meta-ensemble, predicted values ≥ 0.9821 where considered positive (Yes), and those < 0.9 were considered negative (No). Colours in the above matrices indicate agreement between the predicted and the known associations (blue), or no

agreement (blue). Transparency (alpha) indicates probability of agreement (the more times the two sets agreed, in relation to the total space (of yes or no) the more opaque the matrix cell.

“This ‘no-preconceptions’ approach enables us to analyse without being restricted by our current incomplete knowledge of the specific biological and molecular mechanisms which govern host- virus permissibility” - This is presented as a strength, but it’s also a big limitation. These predictions are being made with limited information, so the authors need to be careful with the level of confidence they ascribe to their findings.

Our response: We appreciate the reviewer’s view that ‘no-preconceptions’ approach is only presented as a strength. We had detailed the limitations of the approach in the next section but had not made it clear enough that these limitations were also a part of the ‘no-preconceptions’ approach. Consequently, we have lessened the emphasis on our no-preconceptions approach being a ‘great strength’ and linked and clarified the section describing it as such to the limitations in the next section. This highlights the shortfalls of using the more limited information than would be possible if we were focusing only on well-studied viruses/hosts.

Changes – Discussion (line 228-321): By constructing a comprehensive set of similarity learners in each point of view and combining these learners non-linearly (via GBM meta-ensemble) a strength of our analytical pipeline is that it is able to predict potential recombination hosts of coronaviruses without any prerequisite knowledge or assumptions. Our method does not make assumptions about which parts of the coronavirus or host genomes are important, or integration of receptor (e.g. ACE2) information, or focusing on certain groups of hosts (e.g. bats or primates). This ‘no-preconceptions’ approach enables us to analyse without being restricted by our current incomplete knowledge of the specific biological and molecular mechanisms which govern host-virus permissibility. Current restrictions include lack of sequencing, annotation and expression analysis of receptor (e.g. ACE2) in the vast majority of hosts, uncertainty over the receptor(s) utilised by many coronaviruses, and knowledge of other factors leading to successful replication once the virus has entered the host cell. Whilst some of these details are known for a very limited number of well-studied hosts and coronaviruses, they are not for the vast majority, consequently, a study aiming for breadth of understanding across all mammalian hosts and coronaviruses is unable to utilise these limited data. Despite our ‘no-preconceptions’ approach having this distinct advantage, it is also a limitation of the predictions. As discussed in the next section, our predictions are consequently reliant upon a more limited set of information due to the breadth of the work. Where some data are available for a small subset of coronaviruses or their hosts (e.g. pathogenicity, virus titre), these data are not useable in this study as they do not exist for the vast majority of hosts/viruses.

We acknowledge certain limitations in our methodology, primarily pertaining to current incomplete datasets in the rapidly developing but still understudied field:

1) The inclusion only of coronaviruses for which complete genomes could be found limited the number of coronaviruses (species or strain) for which we could compute meaningful similarities, and therefore predict potential hosts. The same applies for our mammalian species – we only included mammalian hosts for which phylogenetic, ecological, and geospatial data were available. As more data on sequenced coronaviruses or mammals become available in future, our model can be re-run to further improve predictions, and to validate predictions from earlier iterations.

2) Virological knowledge of understudied coronaviruses and their host interactions. For the vast majority of observed virus-host associations it is unknown if these hosts are natural, intermediate, or ‘dead-end’ hosts. Also unknown are more clinical traits of the infections in the overwhelming majority of associations, such as: pathogenicity, likelihood of infection, virus titre during infection, duration of infection, etc. knowledge of all of these factors could greatly add to our ability to assess ‘likeliness’ of homologous recombination, however, the available data are too limited for a study with the breadth of interactions we characterise here, and hence were unable to be included.

The findings don't show that the number of interactions and potential recombination hosts are underestimated. They predict this, but these predictions need to be validated before we can say with confidence that host interactions were previously underestimated.

Our response: we changed mentions of show (and similar) to predict (and similar) throughout the manuscript (including the examples highlighted by the reviewer below).

Discussion - "potential hosts" should be replaced or qualified with "predicted hosts", as potential could refer to a known host that could potentially be infected and facilitate recombination, rather than a predicted host that has no empirical evidence to support it yet. E.g. "indicating that observed data are missing 31.5-fold of the total number of potential recombination hosts" should be changed to "indicating that observed data are predicted to be missing 31.5-fold of the total number of potential recombination hosts"

Our response: done. Sentence changed accordingly.

"Furthermore, our results show that the number of viruses which could potentially recombine even within these known hosts has been significantly under-ascertained, indicating that there still remains significant potential for further novel coronavirus generation in future from current known recombination hosts." versus

"Furthermore, our results suggest that the number of viruses which could potentially recombine even within these known hosts may have been significantly under-ascertained, which would mean that there still remains significant potential for further novel coronavirus generation in future from current known recombination hosts." accordingly

Our response: done. Sentence changed accordingly.

Reviewers' Comments:

Reviewer #4:

Remarks to the Author:

I think the interpretation of probabilities given by ML algorithms is a thorny issue, and calling them probabilities when we know training data are heavily biased and incomplete may be misleading. I'm not sure I agree with the logic that a score of 0.5 means a 50% probability of a true interaction, since the model was not trained on which species do and don't interact, it was trained on observed and unobserved interactions, so that is all it can learn to predict. Nevertheless, I think the adjustment of language in the paper to communicate that these are all predictions sufficiently addresses the uncertainty here. I agree that post-hoc confirmation of some of these associations is indeed a positive sign.

I am very happy with the added discussion and feel that now the value and limitations of the findings presented here is well-communicated.